# The INSIEME seismic network: a research infrastructure for studying induced seismicity in the High Agri Valley (southern Italy)

Tony Alfredo Stabile[1], Vincenzo Serlenga[1], Claudio Satriano[2], Marco Romanelli[3], Erwan Gueguen[1], Maria Rosaria Gallipoli[1], Ermann Ripepi[1], Jean-Marie Saurel[2], Serena Panebianco[1,4], Jessica Bellanova[1], Enrico Priolo[3]

[1]Istituto di Metodologie per l'Analisi Ambientale, Consiglio Nazionale delle Ricerche, Tito (PZ), 85050, Italy
[2]Université de Paris, Institut de physique du globe de Paris, CNRS, UMR 7154, F-75238 Paris, France
[3]Centro di Ricerche Sismologiche, Istituto nazionale di oceanografia e di geofisica sperimentale, Sgonico (TS), 34010, Italy
[4]Università degli Studi della Basilicata, Dipartimento di Scienze, Potenza (PZ), 85100, Italy

*Correspondence to*: Tony Alfredo Stabile (tony.stabile@imaa.cnr.it)

**Abstract.** The High Agri Valley is a tectonically active area in southern Italy characterized by high seismic hazard related to fault systems capable of generating up to M=7 earthquakes (i.e., the 1857 Mw 7 Basilicata earthquake). In addition to the natural seismicity, two different clusters of induced microseismicity were recognized to be caused by industrial operations carried out in the area: (1) the water loading and unloading operations in the Pertusillo artificial reservoir; (2) the wastewater disposal at the Costa Molina 2 injection well. The twofold nature of the recorded seismicity in the High Agri Valley makes it an ideal study area to deepen the understanding of driving processes of both natural and anthropogenic earthquakes and to improve the current methodologies for the discrimination between natural and induced seismic events by collecting high-quality seismic data. Here we present the dataset gathered by the INSIEME seismic network that was installed in the High Agri Valley within the SIR-MIUR research project INSIEME (INduced Seismicity in Italy: Estimation, Monitoring, and sEismic risk mitigation). The seismic network was planned with the aim to study the two induced seismicity clusters and to collect a full-range of open-access data to be shared with the whole scientific community. The seismic network is composed of eight stations deployed in an area of 17 km × 11 km around the two clusters of induced microearthquakes and it is equipped with triaxial weak-motion broadband sensors placed at different depths down to 50 m. It allows to detect induced microearthquakes, local/regional earthquakes, and teleseismic events from continuous data streams transmitted in real-time to the CNR-IMAA Data Centre. The network has been registered at the International Federation of Digital Seismograph Networks (FDSN) with code 3F. Data collected until the end of the INSIEME project (2019-03-23) are already released with open-access policy through the FDSN webservices and are available from IRIS DMC (https://doi.org/10.7914/SN/3F_2016; Stabile et al., 2016). Data collected after the project will be available with permanent network code VD (https://doi.org/10.7914/SN/VD) as part of the High Agri Valley geophysical Observatory (HAVO), a multi-parametric network managed by the CNR-IMAA research institute.

## 1 Introduction

Anthropogenic seismicity has been documented since the 1920s when the subsidence due to the exploitation of the Goose Creek oil field (USA) was responsible of felt earthquakes (Pratt and Johnson, 1926). Nowadays it is commonly accepted that the term induced seismicity is synonymous with anthropogenic seismicity; therefore, in this paper the two terms are considered as interchangeable.

Considering the strong socioeconomic impact of induced seismicity (for a complete review see: National Research Council, 2013; Ellsworth, 2013; Grigoli et al., 2017; Foulger et al., 2018; Keranen and Weingarten, 2018; Lee et al., 2019), the current research in this field has twofold importance: a) from a social and economic point of view it is useful for addressing the range of issues related to the induced seismicity, including the development of specific "best practice" protocols, monitoring strategies and traffic light systems, the correct definition of the associated hazard and risk, and the discrimination between natural and induced seismicity; b) from a pure scientific point of view the research is fundamental for better understanding the processes involved in earthquake generation, the interactions among rock, faults, and fluids as a complex system, and how perturbations of the stress field, even of small size, may affect the stability of faults over time. In order to achieve these two main goals, adequate monitoring networks should be deployed in the study area with the aim to obtain accurate earthquake locations and to lower the completeness magnitude for generating huge microseismic catalogues (Grigoli et al., 2017).

On these grounds, in 2016 a dense seismic network (named INSIEME) was installed in the High Agri Valley (hereinafter HAV), a NW-SE trending intermontane basin formed during the Quaternary age along the axial zone of the southern Apennines thrust belt chain of Italy (Patacca and Scandone, 1989). Indeed, the area hosts energy technologies that cause two clusters of anthropogenic seismicity. More specifically, one of the two clusters (cluster A in Fig. 1) is continued-reservoir induced seismicity ($Ml \leq 2.7$) linked to the seasonal water level fluctuation of the artificial Pertusillo Lake (Valoroso et al., 2009; Stabile et al., 2014a, 2015; Telesca et al., 2015; Vlček et al., 2018); the other cluster (cluster B in Fig.1) is fluid-injection induced seismicity ($Ml \leq 2$) due to the disposal of the wastewater produced during the exploitation of the biggest onshore oil and gas field in west Europe at the Costa Molina 2 (CM2) injection well (Stabile et al., 2014b; Improta et al., 2015, 2017; Wcisło et al., 2018). Furthermore, HAV is one of the areas of Italy with the highest seismic hazard with an expected maximum acceleration (referred to average hard ground conditions) for an exceedance probability of 10% in 50 years within 0.25 and 0.275 g according to the national reference seismic hazard model (Gruppo di Lavoro MPS, 2004). Indeed the Italian historical seismicity catalogue CPTI11 (Rovida et al., 2011) reports seven earthquakes with $Mw \geq 4.5$ in the HAV, including the 1857 Mw 7.0 Basilicata earthquake (Mallet, 1862; Burrato and Valensise, 2008) which was one of the most destructive historical earthquakes in Italy with 11,000 casualties and extensive damage throughout Basilicata, Campania, Apulia, and Calabria Regions. It has been also been estimated from GPS velocity and strain rate field data (D'Agostino, 2014) that the extensional opening in the axial part of southern Apennines is about 3 mm/yr.

The INSIEME seismic network was designed and developed in the framework of the research project INSIEME (INduced Seismicity in Italy: Estimation, Monitoring, and sEismic risk mitigation), which was funded in 2015 by the SIR (Scientific

Independence of young Researchers) program of the Italian Ministry of Education, Universities and Research (MIUR) and ended on 2019-03-23. Two Italian test sites have been selected for the project's research activities: (a) the Collalto area in the municipality of Susegana (Veneto Region, northeastern Italy), site exploited by Edison Stoccaggio S.p.A. for the storage of the natural gas; (b) the High Agri Valley (Basilicata Region, southern Italy) hosting the biggest on shore oil field in west Europe, managed by Eni S.p.A., and the Pertusillo water reservoir. The Collalto site is already being monitored since 2012 by a dense network of 10 seismic stations and one permanent GNSS geodetic station (https://doi.org/10.7914/SN/EV), which was the first Italian network providing data with open-access policy (Priolo et al., 2015).

In this paper, we present the INSIEME seismic network and a detailed description of the acquired data which are released through open access. The broadband seismic network has been registered at the International Federation of Digital Seismograph Networks (FDSN, http://www.fdsn.org, last access: January 2020) with code 3F (https://doi.org/10.7914/SN/3F_2016). Section 2 details the seismic network from its layout to the acquisition, transmission and preliminary processing of data. Section 3 is focussed on the description of acquired seismic signals from the data quality of continuous data streams to the waveforms of recorded seismic events. Section 4 provides information on data availability. Finally, our discussions and conclusions are reported in Section 5.

## 2 The INSIEME seismic network

The INSIEME seismic network has been designed to achieve two main purposes: a) to study the seismic processes related to the occurrence of events belonging to the two clusters of anthropogenic seismicity; b) to provide the scientific community with new open-access high-quality seismic data for studying such phenomenon and for developing methodologies useful to discriminate between natural and anthropogenic events. The following points provide details of the network from its layout to the data acquisition and processing.

### 2.1 Seismic network layout

The INSIEME network is composed of eight stations covering an area of about 17 km × 11 km, organized in two groups of four stations around each of the two clusters of anthropogenic events (red circles in Figure 1). All the stations are equipped with broadband sensors installed in non-toxic PVC (polyvinyl chloride) casings at different depths down to 50 m. Network layout and design definition was driven by several constraints, hereafter summarized:

- seismic stations must be deployed around the two clusters of induced events with as uniform as possible azimuthal distribution;

- taking into account that the studied clusters are characterized by shallow events of about 4-5 km focal depth (Serlenga and Stabile, 2019), the epicentral distance of the closest station must be less than the focal depth of events belonging to such seismicity clusters and the average distance between stations should not exceed twice the event depth (Havskov et al., 2012); these two conditions allow for a good control of event depth estimation;

- as the studied clusters are characterized by shallow events, of about 4-5 km focal depth (Serlenga and Stabile, 2019), the average distance between stations should not exceed twice that depth (Havskov et al., 2012) to correctly estimate the event depth;

- high quality sites, possibly on hard bedrock and therefore without local ground effects, should be selected;

- recommended station locations must be as far as possible from strong noise sources such as main roads, town centers, industrial and quarry activities which are largely diffused in the HAV;

- station sites must be accessible for drilling operations;

- in areas belonging to the National Park "Val d'Agri - Lagonegrese", which covers large portions of the HAV territory (see file "INSIEME-network.kmz" provided in the Supplement), it is not possible to drill boreholes;

- stations sites should be covered by 3G mobile communication link;

- seismic stations should guarantee continuous data acquisition in all weather conditions, even in the winter season where the snow coverage could reach up to 1.5-2.0 m thickness for a couple of weeks;

- with the aim to provide an effective added value to the seismic monitoring of the HAV, station locations should not overlap existing stations of operating public and private seismic networks.

We performed seismic ambient noise measurements and geological surveys in order to find the most suitable sites according to these constraints, and we verified the access to the site (also for drilling operations of the shallow boreholes), data transmission conditions and unexpected potential sources of local noise. We evaluated the network performances following the approach proposed by Stabile et al. (2013) by considering different configurations of the potential sites that have met as many constraints as possible. The final network configuration is reported in Figure 1. It is worth noting that the minimum

distances between stations (cyan triangles in Figure 1) ranges between 2.7 km (INS6 and INS7 stations) and 5.4 km (INS1 and INS4), the distance of the closest station to each cluster is less than 4-5 km (the focal depth of induced events) and the INSIEME stations do not overlap stations belonging to other public (orange triangles in Figure 1) and private (yellow triangles in Figure 1) seismic networks. Only INS8 station falls in the National Park "Val d'Agri - Lagonegrese" area and, therefore, the sensor of this station was installed on the surface.

**2.2 Seismic stations**

Considering that the main target of the INSIEME network is to detect and locate the anthropogenic microseismicity in the HAV (Ml ≤ 2.7), the seismic stations were equipped with triaxial weak-motion broadband sensors: six 0.05-100 Hz and two 0.0083-100 Hz Trillium Compact Posthole (TCPH) seismometers. The data-loggers are Centaur Digital Recorders with a dynamic range of 140 dB. All seismometers and dataloggers are manufactured by Nanometrics Inc. (see Table 1). Continuous

acquisition of digital waveforms is provided by the INSIEME network at 250 Hz sampling rate. This choice allows data acquisition with a Nyquist frequency of 125 Hz (Figure 2), which is greater than the upper frequency bound of the broadband sensors (100 Hz), thus avoiding the application of temporal anti-aliasing filters on the acquired signals and taking advantage of the high frequency boud provided by the sensors useful to capture the full spectra content of small earthquakes. The

amplitude and phase responses of the two versions of broadband sensors are shown in Figure 2: blue curves refer to INS1 station (equipped with a 0.0083-100 Hz TCPH seismometer) and orange curves refer to INSX station (equipped with a 0.05-100 Hz TCPH seismometer). The first installed station of the network was INSX (in operation between 2016-04-01 and 2017-01-24) whereas the other stations have been installed from 2016-09-23 (see Table 2).

With exception of the INS1 station which was initially connected to the electric power grid of the Montemurro Cemetery, power supply for all stations is provided by solar panels and batteries. Each station is equipped with a 270 W solar panel and two 12 V, 100 Ah batteries connected in series to output 24 V, which allow the instruments to work with less current. Solar panels are installed on 2 m high poles in order to prevent snow covering during the winter season (see Figure 3a). The solar charge controller, the two batteries, the power supply circuit, the data-logger and the router are housed in a small cabin (Figure

3b). Corrugated cables allow the passage of sensor cables from the cabin to the borehole (Figure 3b). Each borehole is closed by a manhole (Figure 3b) and the PVC casing is coupled to the soil by cement grout filling the space between the hole and external surface of the PVC casing from the bottom to the surface (Figure 3c). The PVC casing is not in the manhole in order to leave room for installing sensors on the surface (Figure 3d). A 2 m high netting, surrounding an area of about 2.5 m x 2.5 m, protects each station from wild or grazing animals.

The broadband seismometers installed in boreholes are equipped with a coupling system (Figure 3e), developed by the National Institute of Oceanography and Experimental Geophysics of Italy (OGS), which fastens the sensor to the borehole wall. The inclination of each borehole from the surface to the bottom has been measured with an in-place inclinometer (Jewell Instruments, Model 906 Little Dipper). We found that the 5 shallow boreholes of 6 m depth (stations INS2, INS3, INS4, INS5, and INS6) have inclination at the bottom less than one degree and one of the two 50 m deep boreholes deviates 1.6 degrees at

the bottom (station INS1). Concerning the second 50 m deep borehole (station INS7), the inclination of the borehole becomes greater than two degrees at depths greater than 20 m. Indeed, at 14 m depth we measured an inclination of 1.7 degrees, increasing up to 2 degrees between 16 m and 20 m and over 6 degrees beyond 24 m depth. Since the two deeper boreholes host the 0.0083-100 Hz TCPH seismometers which operate with a maximum tilt of 2 degrees, we installed the seismometer of station INS1 at 50 m depth whereas the INS7 one was installed at 14 m depth. Table 2 indicates the sensor depths of each

borehole station.

The seismometers at 6 m depth were installed by a modular non-rotating pipe system developed by OGS in order to control the orientation of the horizontal components (Figure 3d). The non-rotating system consisted of a set of connectable, light and rigid pipes, 3 m long and 50 mm outside diameter, equipped with a mating joint at the velocimeter end, and a reference mark at the top, thus allowing to push the sensor sled and, at the same time, set the correct azimuthal angle. After the installation

was completed we released the joint by lifting and removing the tubes, and the sensor stands undisturbed. For the two seismometers placed at 14 m and 50 m depth, respectively, the orientation of their horizontal components was unknown because of the impossibility to use a longer non-rotating pipe system. In this case we estimated their azimuthal orientation with respect to a reference station by applying a methodology similar to that proposed by Zheng and McMechan (2006), based on the maximization of the cross-correlation among the horizontal traces of adjacent sensor pairs. Of course, for each pair of

adjacent sensors we assume the condition of plane wave approximation which is satisfied if the distance $d$ between sensors is much less than dominant wavelength $\lambda$ of the recorded signal ($d \ll \lambda$). Therefore, the signals recorded by the two sensors must be filtered with a cut-off frequency $f_c \ll V d^{-1}$, with $V$ the lowest seismic velocity of the medium.

For each angle $\theta$ ranging from 0 to 360 degrees with a step size of 0.5 degrees, we computed the normalised cross-correlation

between the North component of the signal recorded by the reference station ($Sr_N$) with the first horizontal component of the signal recorded by sensor with unknown orientation and rotated counter-clockwise by the angle $\theta$ ($Su_1{}^\theta$). In addition, we computed the normalised cross-correlation between the East component of the signal recorded by the reference station ($Sr_E$) with the second horizontal component of the signal recorded by sensor with unknown orientation and rotated counter-clockwise by the angle $\theta$ ($Su_2{}^\theta$). For each angle $\theta$, the maximum value of the cross-correlation between $Sr_N$ and $Su_1{}^\theta$ ($A^\theta$) and between $Sr_E$

and $Su_2{}^\theta$ ($B^\theta$) were retrieved. Then, the sensor orientation with respect to the reference sensor was given by the following Eq. (1):

$$\theta^{BEST} = \theta : \max_{0° \leq \theta \leq 360°} \left( A^\theta B^\theta \right), \tag{1}$$

where $\theta^{BEST}$ is the angle for which the product between $A^\theta$ and $B^\theta$ is maximum. By applying Eq. (1) over $N$ recordings, we obtained $N$ estimates of $\theta^{BEST}$; therefore, we evaluated the weighted arithmetic mean and the weighted standard deviation of

all the $N$ estimates, with $W_i = (A^\theta B^\theta)_i$ the weight of the $i$-th $\theta_i{}^{BEST}$.

For station INS1 we used as reference the station INSX, whose sensor was only 70 m away from the borehole sensor of station INS1. Indeed, both INS1 and INSX were in operation from 2016-10-12 to 2017-01-24 (Table 1 and Table 2). We applied a bandpass filter to the seismic recordings with corner frequencies of 0.1 Hz and 0.5 Hz because we surely satisfy the relation $f_c \ll V d^{-1}$ (i.e. considering that Vs=510 m s$^{-1}$ (Giocoli et al., 2015), and d=$(70^2+54^2)^{0.5}$=88 m) and because in this frequency

range we can use also the seismic ambient noise for determining the rotation angle; in this frequency range, indeed, noise contains the microseismic peak (Longuet-Higgins, 1950) between 4-8 s (0.125-0.250 Hz) which is very coherent. To compute the rotation angle, we used recordings of three earthquakes (Mw 6.5 Central Italy earthquake of 2016-10-30, Mw 5.4 Greece earthquake of 2016-10-15, and Mw 7.9 Papua New Guinea earthquake of 2017-01-22) and seismic ambient noise data of different durations (20 minutes, one hour, and two hours). The final estimate of the rotation angle for the sensor of station

INS1 is 307.8±0.4 degrees counter-clockwise to the North.

For station INS7 we used as reference the station INS1 after its alignment to the North because the two stations are both equipped with the same broadband sensors (Table 1). Since 2017-03-23, when the sensor of INS7 station was installed in the borehole (Table 2), both INS1 and INS7 stations recorded several teleseismic events. We used the surface waves of 12 selected teleseismic events with Mw≥6.9 occurred between November 2017 and August 2018. The distance between stations is d=11

30  km; therefore, we applied a low-pass filter to the seismic recordings with a corner frequency of 0.05 Hz which satisfies the relation $f_c \ll V d^{-1}$ even if we consider a Rayleigh wave speed as low as 2.8 km s$^{-1}$. The final estimate of the rotation angle for the sensor of station INS7 is 43.8±0.3 degrees counter-clockwise to the North.

## 2.3 Seismic data acquisition, data transmission, visualization and preliminary processing

Seismic data are transmitted in real time by 3G mobile system. The modem/router adopted is Teltonika RUT-500; this device is capable of communicating with every 3G Italian mobile network and has also an integrated 4-port RJ-45 10/100Mbps Ethernet switch for the Local Area Network. The mobile telecommunication provider allocates a dynamic public IP address to the WAN-interface of the router; for this reason the system cannot be continuously reached from an external network, as the address may change. Therefore, it has been configured a Dynamic DNS (Domain Name System) whose host name is linked up to the router's dynamic IP address. In this way the end user is able to reach directly each seismic station, both for management and data acquisitions.

The use of the dynamic DNS system instead of a VPN system such as OpenVPN was a technical choice because the latter requires a dedicated server and configuration. The dynamic DNS is also supported by other routers already in our warehouse (typically TP-LINK, which does not support VPN) which can temporally replace a Teltonika in case of failure.

Nanometrics Centaur digital recorder uses a data streaming protocol called SeedLink (https://ds.iris.edu/ds/nodes/dmc/services/seedlink, last access: January 2020). This is a transmission protocol system used to make the data available on the Internet, because based on the "Internet Protocol suite TCP/IP" (Transmission Control Protocol / Internet Protocol) standard.

Since it is not uncommon for routers to encounter problems causing the interruption of the internet connection, each station is equipped with two automatic reboot systems. The first one is integrated inside the router and based on a ping utility: if the system does not ping an external public IP for some time, the router is automatically rebooted. The second one is based on an external programmable time switch which periodically (in our case once a week) unplugs for a few seconds the power supply of the router, thus preventing any software bug that could freeze the Teltonika. When the router restarts, the Centaur data-logger is able to send missing data to the CNR-IMAA (National Research Council of Italy, Institute of Methodologies for Environmental Analyses) Data Centre. Despite these precautions, sometimes data gaps may occur due to prolonged temporary absence of the 3G signal or other minor transmission problems. For this reason, a 16 GB SD memory card is mounted on each Centaur which allows the local storage of about 6 months of data. After a check on data availability using the vertical component data stream of each station for the entire period of operation of the INSIEME seismic network (Figure 4), the available data range between 93.8% (INS6 station) and approximately 100% (INS2, INS3, INS7, and INS8 stations). All the gaps due to transmission problems have been filled by using data saved on each SD memory card. The unfilled gaps are related to a programmed temporary shutdown of a station (e.g., maintenance, firmware update) or to undesired problems occurred to a specific station. As an example, the missing data of station INS6 of about 6.2% (corresponding to a cumulative time of about 58 days out of 940 days) is due to a misconfiguration of the solar charge controller on which the night light function was erroneously activated (during sunshine the power was switched off). The problem was understood and definitively solved on 2017-01-24 at 09:52 UTC. After this configuration correction, the gaps at station INS6 have become comparable to those observed to the other stations of the INSIEME seismic network (Figure 4).

In the CNR-IMAA Data Centre there is a Linux server for data acquisition, storage and processing. The server has been equipped with a hardware RAID controller (redundant array of independent disks), configured as a "RAID 1" disk mirroring, to protect the data in case of drive failure. Our configuration features two 4 TB hard disks (i.e. 8 TB RAW space) in RAID 1 mode, ensuring a N+1 disk redundancy and a 4 TB total storage capacity. This configuration is an optimal choice for

applications requiring high availability. In the future, we will upgrade the system by means of a Network Attached Storage (NAS) in order to store data as well as to enhance the system performance and availability. Furthermore, on this server the TCP/IP-based SeedLink standard compliant SeisComP3 (https://www.seiscomp3.org, last access: January 2020) software runs for seismological data acquisition. It acquires data in real-time from the INSIEME stations and neighbour stations, store them in a miniSEED file structure and it is able to make those data available through various standard protocols: SeedLink for real-

time flow, ArcLink (https://www.seiscomp3.org/doc/applications/arclink.html, last access: January 2020) and FDSN webservices (https://www.fdsn.org/webservices, last access: January 2020) for archived data requests. This SeisComP3 software also holds the stations metadata and an event database. A schematic view of the data flow from the data-logger to the Data Centre is displayed in Figure 5.

A dedicated web-based system, WebObs (Beauducel et al., 2010), is used to plot in near real-time numerical strip-chart (called

"SefraN") of a representative subset of the configured stations. SefraN is used to manually identify any event present in the data (Figure 6, top panels). It is associated to the Daybook, a database of all the events that have been identified in the data, whether they can be located or not, based on the availability of both P- and S-wave pickings. Some regional and global events are prefilled with information gathered from INGV (http://terremoti.ingv.it/webservices_and_software, last access: January 2020) and USGS (https://earthquake.usgs.gov/fdsnws/event/1/, last access: January 2020) FDSN event webservices. When a

new event is identified, the information is sent to the SeisComP3 database (Figure 6). The event is then manually picked and located (Figure 6, bottom panel) with SeisComP3 Origin Locator Viewer (scolv), using the 1-D velocity model from Improta et al (2017). The WebObs Daybook displays the event information collected from the SeisComP3 FDSN webservice. A preliminary catalogue of HAV seismicity from September 2016 to March 2019 has been produced and is available at http://doi.org/10.5281/zenodo.3632419 (Stabile et al., 2020).

## 3 Acquired seismic signals

### 3.1 Data quality in terms of background noise level

One of the most important goal of a seismic network is to provide high quality records of a seismic event from a number of stations as large as possible and with a good azimuthal coverage; therefore, if the seismic noise is high at different sites the benefits of modern equipment with large dynamic range are compromised (Havskov et al., 2012).

It is well known that the background noise is due to several factors like temperature changes, weather conditions and anthropogenic noise. The first two factors generally produce low-frequency noise (<0.05 Hz) whereas the latter usually contains high frequencies (> 1 Hz). In addition there is also the microseismic noise in the range 4-8 s generated by the sea

activity (Longuet-Higgins, 1950). Since most of the broadband sensors of the INSIEME seismic network have flat response in the range 0.05-100 Hz (see Table 1) and the seismic network is primarily designed to observe microearthquakes, the main goal of our sensor installations is to attenuate the anthropogenic noise. Several studies have already focused on the attenuation of such specific kind of noise (Young et al., 1994; Withers et al., 1996) or on the attenuation of the noise over a broader range of

frequencies including both low and high frequency noise (Hutt et al., 2017). The results of such studies indicate that a successful reduction of the noise is achieved by placing seismic instruments at depth within a rock layer. Indeed, surface layers above the rock, which have low seismic wave velocities, tend to trap the anthropogenic noise and produce site amplification effects. Furthermore, installing seismic sensors at depth in PVC casing has been demonstrated to be an effective way to attenuate the diurnal temperature variation (Spriggs et al., 2014), as we did for our stations.

With the aim to evaluate the seismic noise attenuation at depth for our stations we first installed the sensors of each station on the surface for a period of about 6 months and subsequently we moved the sensor inside the PVC casing at depth (Table 2). The only exception is the station INS1 whose sensor was directly installed at 50 m depth because the surface station INSX was in operation at the same site until 2017-01-24 (Table 2). By selecting continuous data streams acquired by INS1 and INSX stations for three days, from 2017-01-05 to 2017-01-07, characterised by high natural and anthropogenic noise level, we

computed the Probabilistic Power Spectral Densities (McNamara and Buland, 2004). Figure 7 show the comparison of the Probabilistic Power Spectral Densities (hereinafter PPSD) obtained for each component of INS1 and INSX stations in the period range 0.01-20 s (frequency range 0.05-100 Hz). The colour palette indicates the probability (in percentage) to have a certain noise level as a function of the period. The two grey lines in each panel indicate the New High and Low Noise models, respectively, obtained by Peterson (1993) which are used as reference. The two horizontal components of INS1 station (CH1

and CH2, according to the SEED channel naming standard; https://ds.iris.edu/ds/nodes/dmc/data/formats/seed-channel-naming, last access: January 2020) are compared with the two horizontal components of INSX station (CHE and CHN) and the vertical components (CHZ) of the two stations are compared to each other. It is possible to observe that the noise level is less widespread at 50 m depth than at surface and that for periods below 1 s (frequencies above 1 Hz) we have a reduction of the noise level of about 10 dB on average and up to 20 dB. In Figure 7, periods above 20 s (frequencies below 0.05 Hz) are

highlighted in grey because in such period range it is not possible to compare the PPSD of the two stations since only the sensor of INS1 stations has flat response up to 120 s (see Table 1) and, therefore, only its PPSD is significant.

We computed also the PPSD on continuous data streams acquired by INS1, INS2, and INS4 stations from 2017-12-26 to 2017-12-30, a period again characterised by high natural and anthropogenic noise level. It is possible to note (Table 2) that INS2 and INS4 stations are equipped with sensors installed at 6 m depth. Figure 8 shows the comparison among the PPSD obtained

for the horizontal and vertical components of each station in the period range 0.01-20 s (the sensors of stations INS2 and INS4 are 20s-100Hz Trillium Compact Posthole). In this case we do not observe a significant difference of PPSD between a sensor installed at 50 m depth (as for INS1 station) and a sensor installed at 6 m depth (as for INS2 and INS4 stations), hence we can argue that installing a sensor at 6 m depth is enough to have a noise reduction in the period range 0.01-20 s similar as when installing a sensor at 50 m depth.

In order to better understand how the installation of sensors in PVC casing at least 6 m depth is an effective solution for the seismic noise attenuation, we compared spectrograms over long-time continuous data streams (41 days from 2017-03-02 to 2017-04-11) acquired by the two seismic stations INS5 and INS6. As evinced in Table 2, the broadband sensor of INS5 station was installed at 6 m depth during the whole period of observation; on the other hand, the broadband sensor of INS6 station

was first placed on surface until 2017-03-22 and then moved in the shallow borehole at 6 m depth. Figure 9 shows the comparison of spectrograms at the two stations over the whole investigated period. The noise attenuation of about 20 dB at station INS5 with respect to station INS6 before 2017-03-22 is very clear, particularly for the two horizontal components, but the noise levels are comparable in the period of time when both stations have their sensors installed at depth. After 2017-03-22 it is possible to observe that the high frequency (> 1 Hz) day-night succession of INS5 station is a little bit more pronounced

than the day-night succession of INS6 station because the former is closer to the urban area of Sarconi town than the latter. Finally, it is interesting to observe as expected the increase of the microseismic noise in the range 4-8 s generated by the sea activity during storms (e.g., in the period 06-09 March 2017 as effects of a strong Mistral event in the Tyrrhenian Sea); this phenomenon is masked by the high noise level when the sensors are placed on surface.

Finally, for a comprehensive analysis of the noise level at the different investigated depths, we computed the PPSD over the

entire period of operation of the network for all components of all stations. Figure 10 displays the median values of PPSD for the vertical components (channel CHZ) of sensors installed at different depths and locations. It is worth noting that for frequencies above 1.5 Hz (periods below 0.6 s) the median curves obtained for sensors installed on surface (black lines in Figure 10) are generally 10 dB higher than the median curves obtained for sensors installed at depth (blue, green and red curves in Figure 10). The PPSD computed for each component of each individual station at different depths of the sensor are shown

in the Supplement (Figures S1-S8), which also show with black curves the 5[th], the 50[th] (median), and the 95[th] percentiles. The PPSD functions computed over the whole period of operation of the network confirm that the noise level is more widespread when the sensor is installed on surface with respect to the installation in shallow boreholes (see 95[th] percentile curves in Figures S1-S8) and that there is no significant reduction of the noise level for installation of sensors between 6 m and 50 m depth.

### 3.2 Data quality in terms of local ground effects

Local seismic amplifications due to sensor installation on soft ground can greatly affect spectral analyses of low and moderate earthquakes, the broad-band recording can be useless and the short period signals may be unrepresentative (Havskov et al., 2012). The absence of meaningful site effects was beforehand assessed for properly choosing the future locations of each seismic station of the INSIEME network. In order to check the validity of our choice and the quality of seismic signals a further assessment of the negligible site effect on recorded data has been carried out.

To this purpose earthquake data have been selected from the preliminary catalogue of HAV seismicity (http://doi.org/10.5281/zenodo.3632419; Stabile et al., 2020). With the aim to have more accurate locations, the events have been relocated by means of NonLinLoc code (Lomax et al., 2000) in a 3-D velocity model of the area (Serlenga and Stabile, 2019), allowing us to better distinguish three different categories of seismic events:

a) injection-induced earthquakes (IIE hereinafter), whose epicenters belong to the cluster B located NE of the Pertusillo lake and close to the CM2 injection well (see Figure 1). We also increased the number of IIEs by using a template-matching algorithm based on the cross-correlation processing for single station data proposed by Roberts et al. (1989). In this way we were able to use 164 injection-induced earthquakes;

b) reservoir-induced earthquakes (RIE hereinafter), belonging to the cluster A located SW of the lake (see Figure 1), for a total number of 56 events;

c) local earthquakes (LE hereinafter) located in the HAV. In particular, only events with a magnitude greater than 1.5 were selected, for a total number of 33 events.

In addition to earthquake data, five hours of seismic noise data (SN hereinafter) were extracted in the time window 9:00 –
14:00 UTC of 2018-11-26.

In order to assess the presence of possible local ground effects at the sites where the stations were installed, the selected data were analyzed by applying the Horizontal to Vertical Spectral Ratio technique (HVSR; Nakamura, 1989), both to earthquakes and noise data (HVNSR, where the letter "N" stands for "noise").

To this purpose, each component of earthquake data was cut in time windows which allowed to discard as much as possible
the pre- and post-signal noise. For IIE, RIE and LE data we chose 8 s, 16 s and 32 s wide time windows, respectively, with a corresponding minimum frequency of 0.125 Hz, 0.0625 Hz and 0.03125 Hz. In order to have reliable estimates the spectra were evaluated starting from 10 times the respective minimum frequency (i.e., 1.25 Hz, 0.625 Hz and 0.3125 Hz). The difference in the selected time windows is related to the dissimilar durations of recorded signals of each category of earthquakes. Before computing the Fast Fourier Transform, the mean and the trend were removed from the time series and
signals belonging to any data category were tapered by applying a Tukey window with 5% bandwidth. Then, the computed spectra were smoothed by means of Konno-Omachi function (Konno and Omachi, 1998), with a bandwidth coefficient equal to 40. The HVSR for each earthquake was retrieved from the arithmetic mean of the horizontal amplitude spectral components (EW and NS) over the vertical amplitude spectral component (Z) of the acquired signal, that is:

$$HVSR = \frac{EW+NS}{2Z} \ .$$

(2)

Finally, the average HVSR for each station and earthquake category was computed, along with the ±1σ (one standard deviation). The choice of performing such an analysis on different types of earthquakes, characterized by a heterogeneous location in space, was related to look for possible source and directivity effects on the consequent HVSR measurements.

The 5 hour long SN data, on the other hand, were cut in 130 s wide non-overlapping time windows, which spanned the total temporal extension of the recording, providing a total number of 138 signals with a spectral resolution of 0.007 Hz. The
retrieved time series were processed in an analogous way to the one described before for earthquake data by means of the Geopsy software (Geopsy project; http://www.geopsy.org, last access: January 2020). For each time window and station, the HVNSR was retrieved, taking into account that the horizontal spectrum was computed as the squared average of the two horizontal (EW and NS) components:

$$H = \sqrt{\frac{EW^2 + NS^2}{2}} \,. \qquad\qquad\qquad (3)$$

Finally, the average HVNSR of each station was computed, along with the ±1σ.

The retrieved HVSR and HVNSR are represented in Figure 11. We can assert that most of the stations are characterized by an almost flat H/V curve, independently of the adopted dataset. Furthermore, we separately verified that the choice of an

arithmetic mean or a squared average of the two horizontal components is almost completely irrelevant to the consequent HVSR or HVNSR measurements. The arithmetic average adopted for earthquake data analysis allowed to equally weight possible amplitude peaks related to directivity and azimuthal effects in the HVSR computation. On the other hand, the squared average, which generally overestimates the arithmetic average and which was adopted for analyzing the ambient noise data, did not produce higher amplitude peaks: indeed, the retrieved HVNSR curves are flatter than HVSR ones.

In Figure 11 we observe very low site amplifications, except for INS5 seismic station, where a relevant peak at about 3.5 Hz can be noticed and for INS6 whose HVSR function has a slight amplification between 0.8-3.0 Hz. Some detailed considerations must be done on the results related to station INS1. Previous analyses performed at the same site with ambient noise and earthquake data, by using both a seismometer and an accelerometer located at the surface, and with geological and geophysical (Electrical Resistivity Tomography) surveys allowed to approximately estimate the depth of the bedrock at about 50 m (Giocoli

et al., 2015). Indeed, an amplitude peak between 2 and 3 Hz in the retrieved H/V curves was clearly observable. By looking at Figure 11, this peak is no more present, confirming that the installation of INS1 at 50 m depth allowed us to reach a more rigid (higher acoustic impedance) layer; in addition to it, during perforation operations, it was clearly observed at that depth a sharp lithological change from alluvial deposits to Gorgoglione Formation. At INS1 seismic station, the low amplitude peaks at about 4-5 Hz, 9 Hz and 11 Hz, respectively, are observed in HVSR curves but not in the HVNSR one. We might interpret

these differences as the effect of the down-going earthquake wavefield.

### 3.3 Induced microearthquakes, local earthquakes, and teleseismic events

The continuous data acquisition by the INSIEME seismic network allowed to manually detect, by a visual inspection of recordings through SefraN tool, a total number of 852 local natural and induced earthquakes between September 2016 and March 2019. Then, these were preliminarily located (http://doi.org/10.5281/zenodo.3632419; Stabile et al., 2020) using the 1-

D velocity model by Improta et al. (2017) and Hypo71 algorithm (Lee and Lahr, 1972) embedded in SeisComP3, allowing us to have an initial distinction of the three different categories of seismic events already introduced in the previous section (3.2): IIE, RIE, LE. In order to better locate local event outside the INSIEME network, we build a virtual seismic network composed of eleven seismic stations of the Italian National Seismic Network (FDSN codes: IV, https://doi.org/10.13127/SD/X0FXnH7QfY; MN, https://doi.org/10.13127/SD/fBBBtDtd6q) managed by the Italian National

Institute of Geophysics and Volcanology (INGV), seven stations belonging to the Irpinia Seismic Network (Weber et al., 2007; Stabile et al., 2013; FDSN code: IX), and MARCO station belonging to the Geofon network (FDSN code: GE, https://doi.org/10.14470/TR560404), the latter installed south of Tramutola town in the framework of a joint scientific

cooperation between GFZ-Potsdam and CNR-IMAA institutes; all the stations of the virtual network are located within about 60 km distance from the centre of the INSIEME network.

Here we report the main inferred features for each earthquake category (IIE, RIE and LE), in terms of both seismic signal properties and hypocentral locations:

a) IIE: 43 injection-induced seismic events were manually picked and located. These were identified because belonging to the seismicity cluster induced by fluid-injection operations at the CM2 well (cluster B in Figure 1). The first recording station of such events is INS1, which is the closest receiver, with signals characterized by a difference between the arrival times of S- and P-waves of about 1 s. The average depth retrieved from preliminary event location analyses is about 4.5 km and the maximum recorded local magnitude is Ml = 1.4, related to an induced event occurred on 2018-01-29 at 15:23:10 UTC (Figure

12a), located at about 1.4 km epicentral distance from INS1 station with focal depth of about 3.50 km (Ml=1.4, Lat=40.3182˚N, Lon=15.9842˚E, depth=3.50 km; from http://doi.org/10.5281/zenodo.3632419, Stabile et al., 2020). Most of detected IIE have a magnitude lower or equal than 1; only two of them are characterized by local magnitude 1<Ml≤1.4. Depending on the earthquake energy, the number of stations that recorded the seismic signals changes from a minimum of three for the lowest magnitude event up to 16, taking into account also stations belonging to the virtual seismic network. The waveforms, usually,

have duration less than 7 s at the closest station (INS1) and the highest peak ground velocity amplitude (PGV) measured at that station is about 0.04 mm s$^{-1}$ for the strongest IIE of the catalogue (Figure 12a).

b) RIE: 117 reservoir-induced seismic events were manually picked and located, in the range $0 \leq Ml \leq 1.8$. The P-wave arrivals are usually first detected at either INS5 or INS6 or INS7 seismic station, depending on the earthquake location: indeed, such induced events belong to a wider cluster than IIE one and therefore they are more broadly distributed in the southwestern part

of the seismic network (cluster A in Figure 1). Their average depth is about 4.5 km and the maximum recorded local magnitude is Ml = 1.8, related to an event occurred on 2017-03-02 at 21:39:41 UTC (Figure 12b), located at about 1.9 km epicentral distance    from    INS5    station    (Ml=1.8,    Lat=40.2723˚N,    Lon=15.8840˚E,    depth=4.51    km;    from http://doi.org/10.5281/zenodo.3632419, Stabile et al., 2020). Because of the proximity of the stations around this seismicity cluster, also RIE earthquakes, in a way similar to IIE, are characterized by a difference between S- and P-wave arrival times

of about 1 s at the closest station to the epicenter and short duration, less than 8 s (e.g. see Figure 12b). The recorded seismic event with lowest magnitude was detected by seven stations, whereas the strongest earthquake was recorded by 12 stations. Finally, the highest peak ground velocity amplitude recorded up to now for this earthquake category is about 0.08 mm s$^{-1}$ (Figure 12b).

c) LE: 692 local natural earthquakes were manually picked and located. The main difference with respect to the IIE and RIE

is that they are not clustered, they are characterized by a widespread distribution in the investigated area, and their average hypocentral depth of about 10 km is more similar to the typical depth of Apennines crustal earthquakes. Most of recorded LE are characterized by a local magnitude < 2: only 39 seismic events out of 692 have greater magnitude. Four earthquakes with a magnitude greater than 3, included in a radius of about 40 km from the center of INSIEME seismic network, have been recorded. The strongest event close to the INSIEME network (epicentral distance of 16 km from INS5 and INS6 stations) is a

Mw 3.8 earthquake (from http://cnt.rm.ingv.it/event/17474201, last access: January 2020). The highest peak ground velocity amplitude of more than 3 mm s⁻¹ was recorded at MTSN station, managed by INGV, which was the closest station located at 5 km epicentral distance. The earthquake was recorded by the whole INSIEME seismic network, as well as by all the stations of the virtual network that were in operation that day (event Ml=4.0, Lat=40.3040˚N, Lon=15.7200˚E, depth=12.10 km reported in http://doi.org/10.5281/zenodo.3632419; Stabile et al., 2020); in Figure 13 the vertical components of the 18 stations that have recorded the earthquake are displayed.

Concerning teleseismic events, the INSIEME network was able to record the most energetic earthquakes occurred worldwide in the period in which the analyses have been carried out. In Figure 14, the recordings at INS1 station of seismic waves generated by the Mw 7.6 Chile earthquake of 2016-12-25 are shown. We specifically choose to display the waveforms at INS1 station since it was installed at 50 m depth and it is a 120 s instrument: these elements allowed to clearly see the most important seismic phases generated by the earthquake and by the effects of propagation inside the Earth. Their theoretical arrival times were computed by means of SeisGram2k software (Lomax, 2008) which uses the Preliminary Reference Earth Model (PREM) published by Dziewonski and Anderson (1981). In addition to different seismic phases, in Figure 14 we are able to observe the dispersive character of surface waves, with the lower frequencies, travelling deeper in the Earth and, therefore, faster, arriving at INS1 station before the higher frequencies.

## 4 Data availability

The INSIEME network has been registered at the International Federation of Digital Seismograph Networks (FDSN), which assigned the network code 3F (2016-2019) (https://doi.org/10.7914/SN/3F_2016; Stabile et al., 2016). Open-access policy on these data has been adopted under the license CC BY 4.0. Continuous seismic data are available at IRIS DMC from 2016-04-01 to 2019-03-23 (see Figure 4 illustrating the availability of seismic data for all stations of the network). From IRIS DMC FDSN Web Services (https://service.iris.edu, last access: January 2020) it is possible to download the standard StationXML file (Service Interface "station") of each station of the network which reports comprehensive information of the station including the instrument response, the time series data in miniSEED and other formats (Service Interface "datalesect"), and the time series data availability (Service Interface "availability").

The events preliminarily located with the Origin Locator Viewer (scolv) tool of the SeisComP3 software are available in the "Preliminary catalogue of High Agri Valley seismicity (southern Italy) recorded by the temporary INSIEME network" CSV file (available at http://doi.org/10.5281/zenodo.3632419; Stabile et al., 2020). The KMZ file "INSIEME-network.kmz" (Keyhole Markup language Zipped, which can be viewed using Google Earth), provided in the Supplement, is an interactive extension of Figure 1 showing also the layout of the virtual network used to preliminary locate all the events.

Following the end of the SIR-MIUR INSIEME project (2019-03-23), the temporary INSIEME network is going to be updated as a permanent open-access seismic network under the license CC BY 4.0; therefore, acquired data after 2019-03-23 will be available from the permanent network with code VD (https://doi.org/10.7914/SN/VD).

# 5 Discussion and conclusions

In this paper we have presented the data collected by the INSIEME seismic network deployed in the HAV to support the research activities of the SIR-MIUR INSIEME project with new high-quality seismic data. Beyond the research purposes of the INSIEME project, we have adopted open-access policy on the continuous data streams acquired by the seismic network since the beginning of data acquisition with the aim to share with the whole scientific community data collected in a very interesting area where both natural and induced events are observed. In this sense, the network can be considered as an open-access research infrastructure for studying induced seismicity processes and for developing methodologies for discriminating between natural and induced earthquakes.

All the eight stations of the network are equipped with broadband sensors (Table 1), seven of which are installed in PVC casings at different depths down to 50 m (Table 2). The power supply of stations is provided by solar panels and batteries. Only INS1 station was initially connected to a power line, but on 2018-04-14 at 19:16:57 UTC the electric power grid of the Montemurro Cemetery (which powered the station) presented voltage and current instability, thus causing a significant disturbance on the seismic signal. After several attempts the problem was definitively by-passed on 2018-06-29, between 7:19 and 10:04 UTC by connecting also this station to a power system based on solar panels and batteries. The quality of acquired data has been investigated for each station in terms of both the background noise level and the local ground effects. Our analyses indicate that the installation of sensors in PCV casing at 6 m depth allows a reduction of the noise level up to 20 dB with respect to the noise level recorded at surface and that there is not a significant difference of the noise level recorded between 6 and 50 m depth. Furthermore, all the stations are installed on sites with negligible site amplification, except station INS5 where a relevant amplification is observed at about 3.5 Hz (Figure 11).

Between September 2016 and March 2019 we have preliminarily located 852 local natural and induced earthquakes (http://doi.org/10.5281/zenodo.3632419; Stabile et al., 2020): 43 events (Ml≤1.4) are classified as IIE, 117 events (Ml≤1.8) are classified as RIE, and 692 events (Ml≤4.2) are LE.

The availability of a continuous data stream will give the advantage to apply robust automated data analysis procedures for earthquake detection and location such as master-event waveform stacking method (Grigoli et al., 2016), the multiband array detection and location method (Poiata et al., 2016), single station (e.g., Roberts at al., 1989) or array (e.g., Gibbons and Ringdal, 2006) template matching algorithms, or to develop and test new algorithms. As an example, we are developing a single station template-matching algorithm and we performed a first test by using some IIE recorded at INS1 station as event templates, obtaining additional 135 detections of IIE. In this way it is possible to lower the detection threshold of the seismic network and, consequently, to decrease the magnitude of completeness which leads to the production of a larger microseismic catalog.

These new seismicity data, hopefully incremented with recordings coming from the stations of the virtual network, could be used for further seismological studies, such as seismic tomography (both elastic and anelastic), comprehensive focal mechanism study of located events (e.g., like the ISC Bulletin published by Lentas et al. (2019)), estimation of source parameters of each individual event, detailed earthquake locations to study the space-time evolution of seismicity and for fault

imaging, and seismic hazard analyses for a better comprehension of the seismic potential of the area. Continuous data streams can be used also for site characterization studies of the HAV area even for the installation of new stations; in this case some stations of the INSIEME network, particularly INS4 station, could be used as reference since it has been proven that they are free of site effects (Figure 11). Besides IIE, RIE and LE categories, continuous acquisition allowed to record teleseisms that

occurred worldwide. These data acquired by the broad-band sensors of the INSIEME network could integrate the data collected by the Global Seismographic Network (GNS, https://earthquake.usgs.gov/monitoring/gsn, last access: January 2020) or by the GEOFON global seismological broad-band network (https://geofon.gfz-potsdam.de, last access: January 2020) for real-time global earthquake monitoring or for global seismology studies. The great coherence of teleseismic recordings provided by such a dense seismic network can be used also as an antenna to track the energy radiated by the propagating rupture along a

fault (e.g., Satriano et al., 2014) or, more generally, for seismic array applications (e.g., Gibbons et al., 2008).

In addition to the applications mentioned above, all based on earthquake recordings, continuous data streams provide also large datasets of noise data to be processed for the continuous monitoring of crustal temporal variation of seismic wave speed in the study area (e.g., Clarke et al., 2011) or, alternatively, for obtaining broad-band surface waves dispersion curves (Bensen et al., 2007) which could be adopted for ambient noise tomography (e.g., Shapiro et al., 2005) of the HAV.

Finally, it is important to highlight that the INSIEME seismic network will continue to operate also after the end of the SIR-MIUR INSIEME project (2019-03-23) by becoming an open-access permanent seismic network of the High Agri Valley geophysical Observatory (HAVO) managed by the CNR-IMAA research institute.

**Author contributions.** TAS led the writing of the paper and VS prepared sections 3.2 and 3.3. TAS and CS worked on the

seismic network layout and on the choice of the acquisition system. MR installed sensors in PVC casing at different depths and evaluated boreholes inclination. TAS evaluated sensors orientation and signal quality in terms of noise level. JMS and ER organized the CNR-IMAA Data Center, including real-time data transmission from the remote stations to the Data Center and the installation of software for data processing. EG and TAS performed geological surveys and verified the access to the site, the data transmission conditions and unexpected potential sources of local noise. HVSR and HVNSR analyses where

performed by VS and MRG. TAS, VS, JB and SP manually picked the seismic phases and located the natural and anthropogenic seismic events. All co-authors provided comments which contributed to the paper.

**Competing interests.** The authors declare that they have no conflict of interest.

**Acknowledgements.** The authors would like to thank the mayors of the municipalities that host the seismic stations of the INSIEME network for their authorizations: Senatro Di Leo (Montemurro), Cesare Marte (Sarconi), Antonio Maria Imperatrice (Grumento Nova), Amedeo Cicala (Viggiano), Franco Curto (Armento), and Mario Solimando (Spinoso). Figures 1, 2, 4, 7, 8, 9, 10, and S1-S8 were drawn using Matplotlib (Hunter, 2007) and/or ObsPy (Beyreuther et al., 2010) Python libraries. Figures 12, 13, and 14 were drawn using the Seismic Analysis Code (Goldstein et al., 2003).

**Financial support.** This work has been funded by the INSIEME project of the Italian SIR-MIUR program (grant no. RBSI14MN31).

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

**Tables**

**Table 1: Geographic coordinates and elevation of the INSIEME broadband seismic stations with indication of the sensor type installed at each station (TCP = Trillium Compact Posthole).**

| Station name | Latitude °N | Longitude °E | Elevation (m a.s.l.) | Sensor type |
|---|---|---|---|---|
| INSX | 40.305686 | 15.989105 | 806 | 20s-100Hz TCP |
| INS1 | 40.305790 | 15.988603 | 802 | 120s-100Hz TCP |
| INS2 | 40.342090 | 15.951559 | 1043 | 20s-100Hz TCP |
| INS3 | 40.328033 | 16.034446 | 880 | 20s-100Hz TCP |
| INS4 | 40.278168 | 16.040405 | 652 | 20s-100Hz TCP |
| INS5 | 40.275704 | 15.906211 | 602 | 20s-100Hz TCP |
| INS6 | 40.229581 | 15.887608 | 745 | 20s-100Hz TCP |

| Station name | | | | | |
|---|---|---|---|---|---|
| INS7 | 40.221487 | 15.917465 | 881 | 120s-100Hz TCP | |
| INS8 | 40.241083 | 15.972221 | 882 | 20s-100Hz TCP | |

**Table 2: Position of the broadband sensors during time for each station with the indication of the sensor depth when it is installed in the borehole.**

| Station name | surface | | borehole | | Sensor depth |
|---|---|---|---|---|---|
| | installation | uninstallation | installation | uninstallation | |
| INSX | 2016-04-01 | 2017-01-24 | - | - | - |
| INS1 | - | - | 2016-10-12 | - | 50 m |
| INS2 | 2016-09-23 | 2017-03-22 | 2017-03-22 | - | 6 m |
| INS3 | 2016-08-26 | 2017-03-22 | 2017-03-22 | - | 6 m |
| INS4 | 2016-08-26 | 2017-03-22 | 2017-03-22 | - | 6 m |
| INS5 | 2016-08-26 | 2016-10-13 | 2016-10-13 | - | 6 m |
| INS6 | 2016-08-26 | 2017-03-22 | 2017-03-22 | - | 6 m |
| INS7 | 2017-03-02 | 2017-03-23 | 2017-03-23 | - | 14 m |
| INS8 | 2017-03-02 | - | - | - | - |

**Figures**

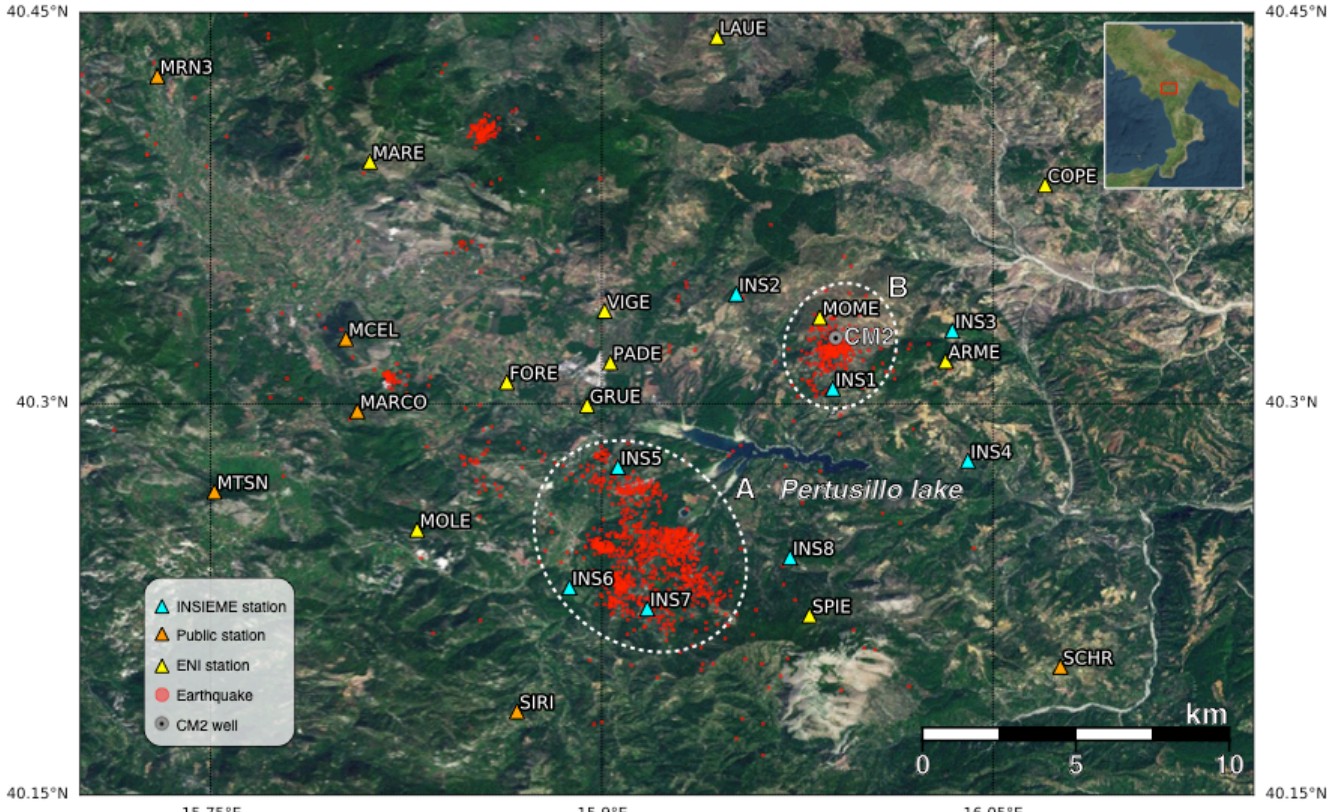

**Figure 1: Layout of the INSIEME seismic network in the High Agri Valley. Cyan triangles represent the 8 broadband seismic stations of the network. Yellow and orange triangles represent stations belonging to private (i.e. the Eni Company) and public seismic monitoring networks, respectively. The CM2 injection well is depicted with a black dot inside a grey circle. Natural and anthropogenic earthquakes are represented with red circles (2002-2012 seismicity from Serlenga and Stabile (2019)). Anthropogenic seismicity is classified as continued reservoir (clusters A) and fluid-injection (cluster B) induced seismicity. The map was drawn using Matplotlib Python library (Hunter, 2007) which incorporates the ArcGIS REST Services freely available at http://server.arcgisonline.com/arcgis/rest/services.**

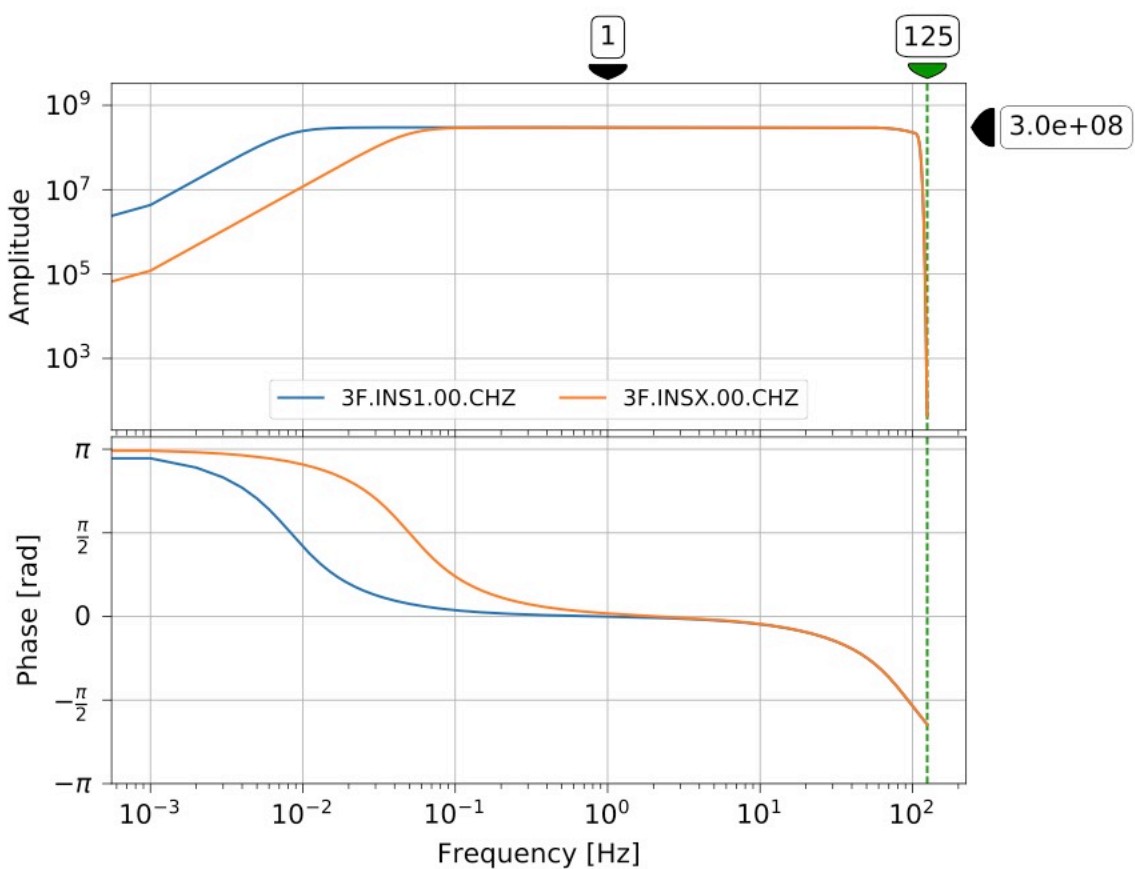

**Figure 2: Amplitude and phase response curves for INS1 station (blue curves), equipped with a 0.0083-100 Hz TCPH seismometer, and for INSX station (orange curves), equipped with a 0.05-100 Hz TCPH seismometer. The sensor sensitivity at 1 Hz is also reported. Vertical dotted green line indicates the Nyquist frequency of 125 Hz.**

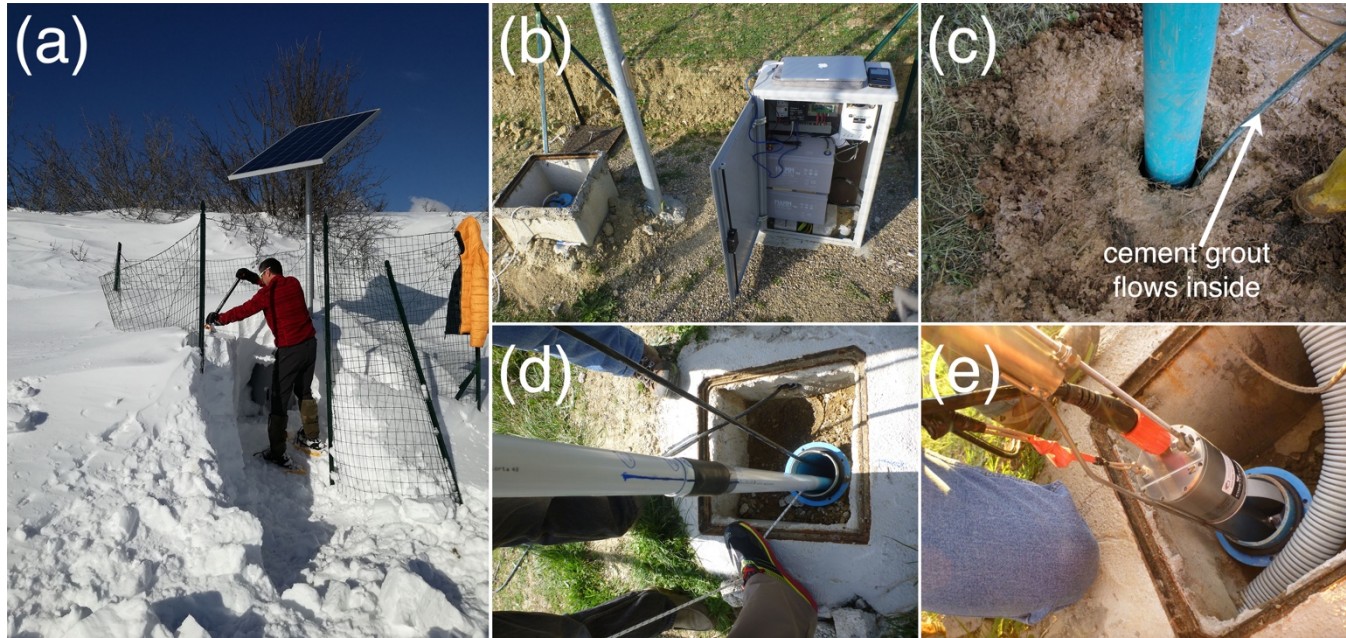

**Figure 3: Details of a typical seismic station of the INSIEME network: (a) the solar panel is installed on a pole of 2 m height in order to prevent that it is covered by snow during the Winter season; (b) all the instruments of a station are housed in a small cabin which is connected to the borehole where the seismometer is installed inside a PVC casing; (c) the PVC casing is coupled to the soil by using a cement grout; (d) the PVC casing is not centered in order to leave space for installing sensors on the surface and seismometers placed at 6 m depth are installed by using a non-rotating pipe system; (e) all the broadband seismometers installed in boreholes are equipped with a coupling system.**

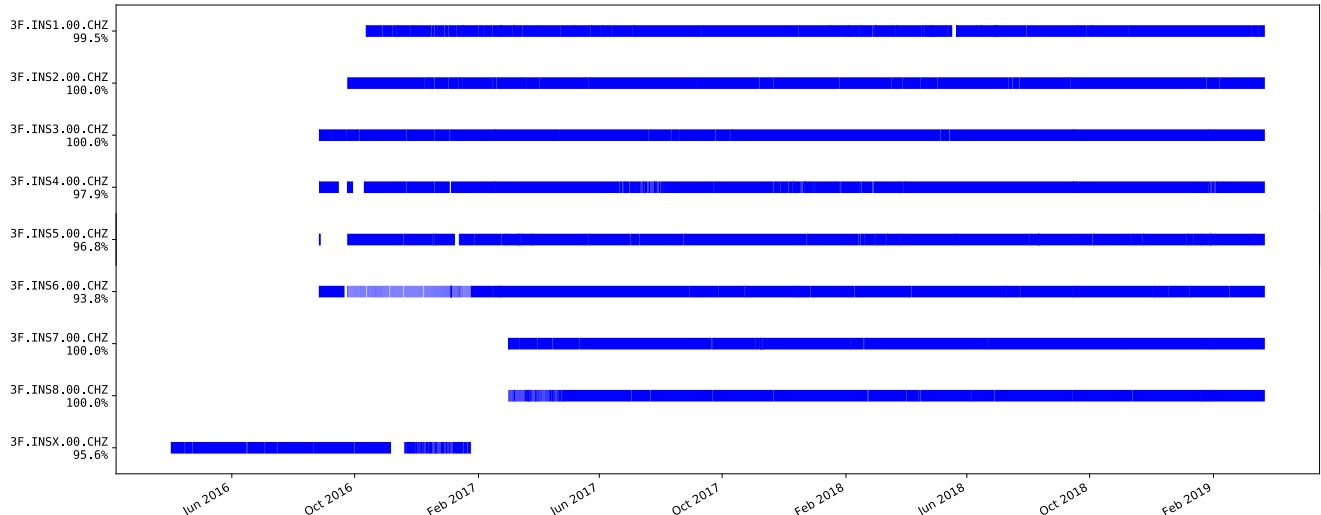

**Figure 4: Data availability (blue lines) for all stations for the entire period of operation of the INSIEME seismic network. The analysis has been performed on the vertical component data stream of each station (CHZ channels). The percentage of data availability is reported below each station name.**

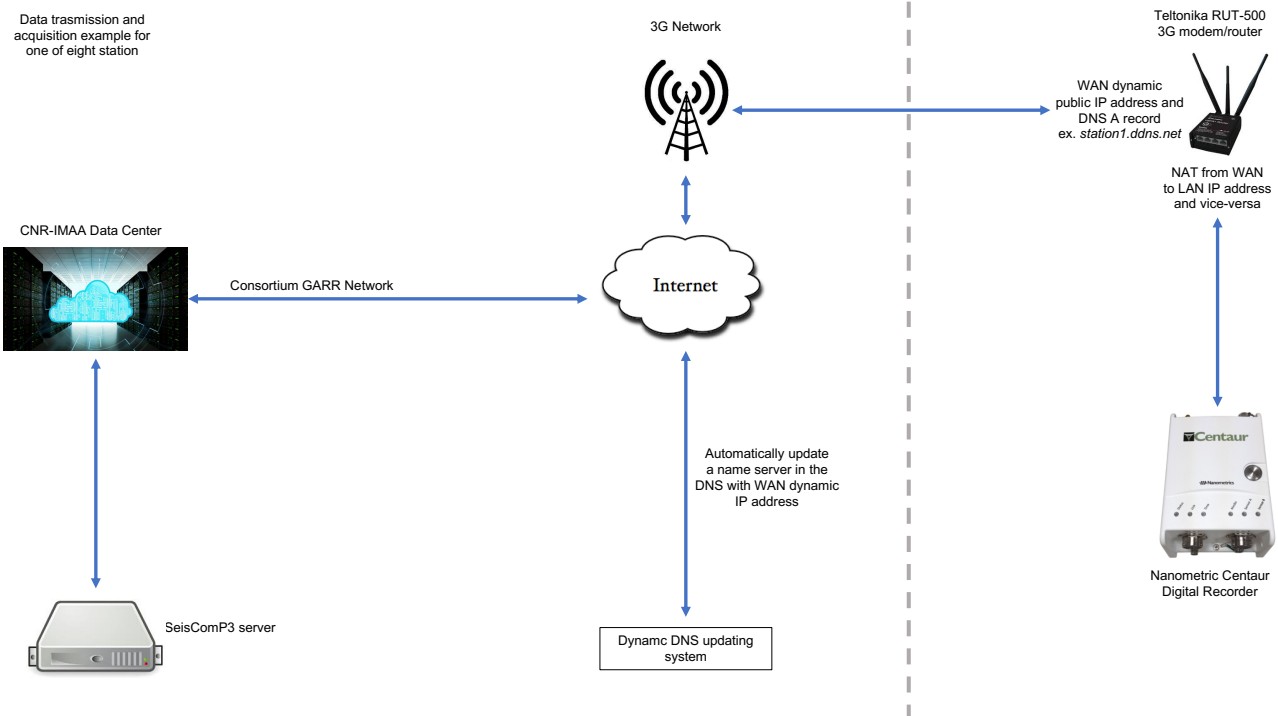

**Figure 5: Schematic view of the data flow from the data-logger of a remote station to the CNR-IMAA Data Centre.**

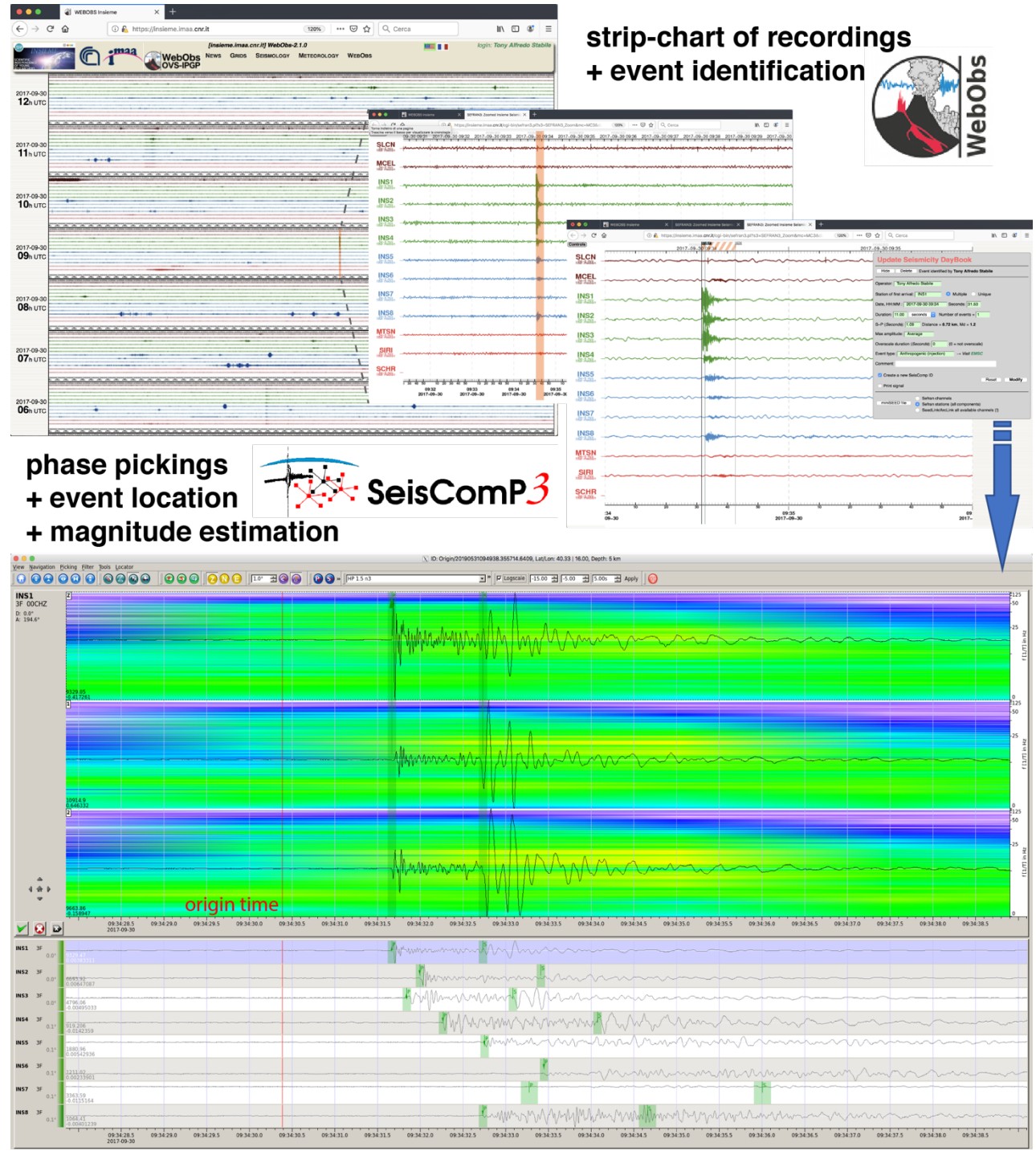

**Figure 6: Tools implemented for visualization and processing of acquired seismic data. The WebObs system (top panels) is used to plot in near real-time strip-chart of recordings at configured stations any event present in the data. When an event is identified, the information is sent to the SeisComP3 database for the manual phase picking and event location (bottom panel) through the Origin Locator Viewer tool (scolv).**

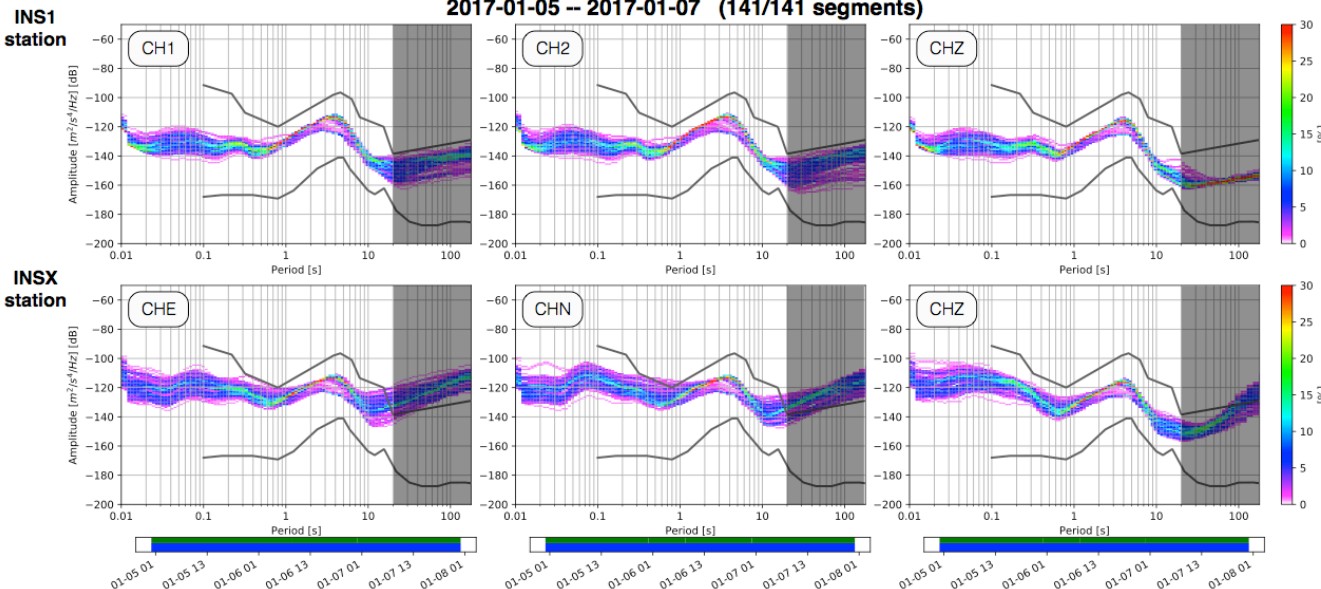

**Figure 7: Probabilistic Power Spectral Densities (PPSD) computed for each component of station INS1 (top panels), with sensor installed at 50 m depth, and station INSX (bottom panels), with sensor installed at surface. The colour palettes on the right indicate the probability (in percentage) to have a certain noise level. The two grey curves in each panel indicate the New High and Low Noise models, respectively, obtained by Peterson (1993). Below all PPSD panels there is visualized the data basis: the top row is coloured in green for available data and in red (not in this case) for eventual gaps in streams. The bottom row is coloured in blue if the single PSD measurement is included in the PPSD computation. Periods above 20 s are highlighted in grey because in such period range it is not possible to compare the PPSD of the two stations (only the sensor of INS1 stations has flat response up to 120 s).**

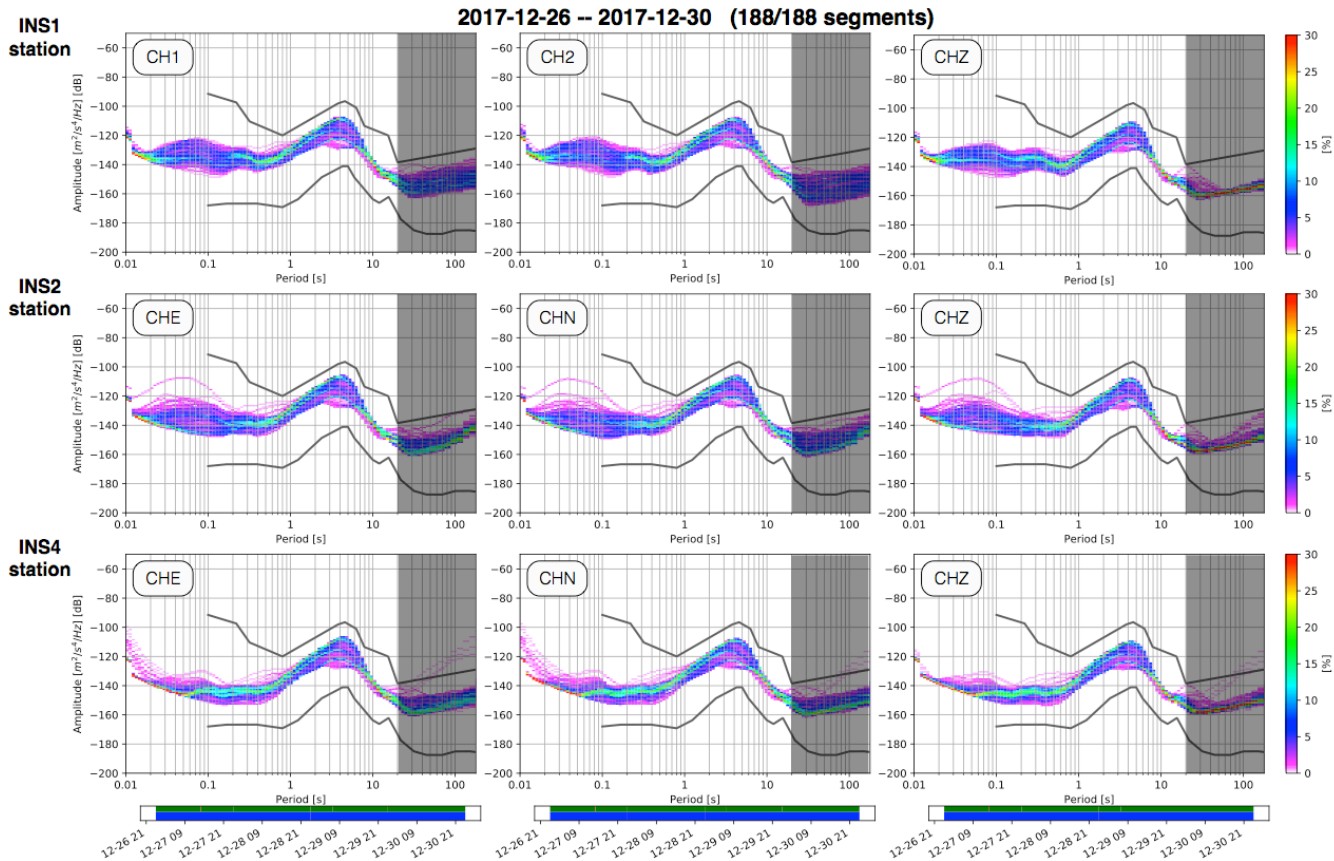

**Figure 8: Same type of comparison as Figure 7 but among stations INS1 (sensor installed at 50 m depth), INS2 and INS4 (respective sensors both installed at 6 m depth).**

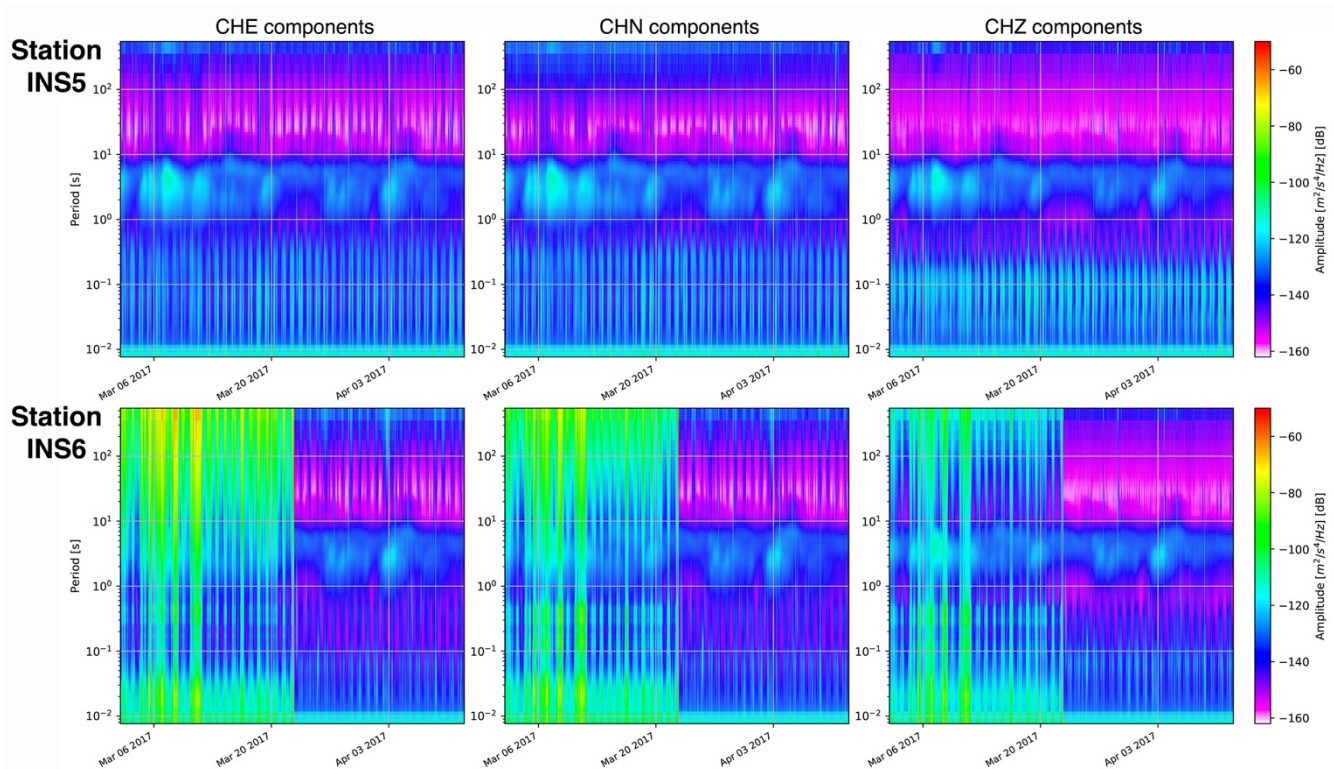

**Figure 9: Spectrograms of each component of station INS5 (top panels) and INS6 (bottom panels) computed over continuous data streams of 41 days (from 2017-03-02 to 2017-04-11). The sensor of station INS5 was installed at 6 m depth for the whole period of observation whereas the sensor of station INS6 was first installed on surface and then moved at 6 m depth on 2017-03-22.**

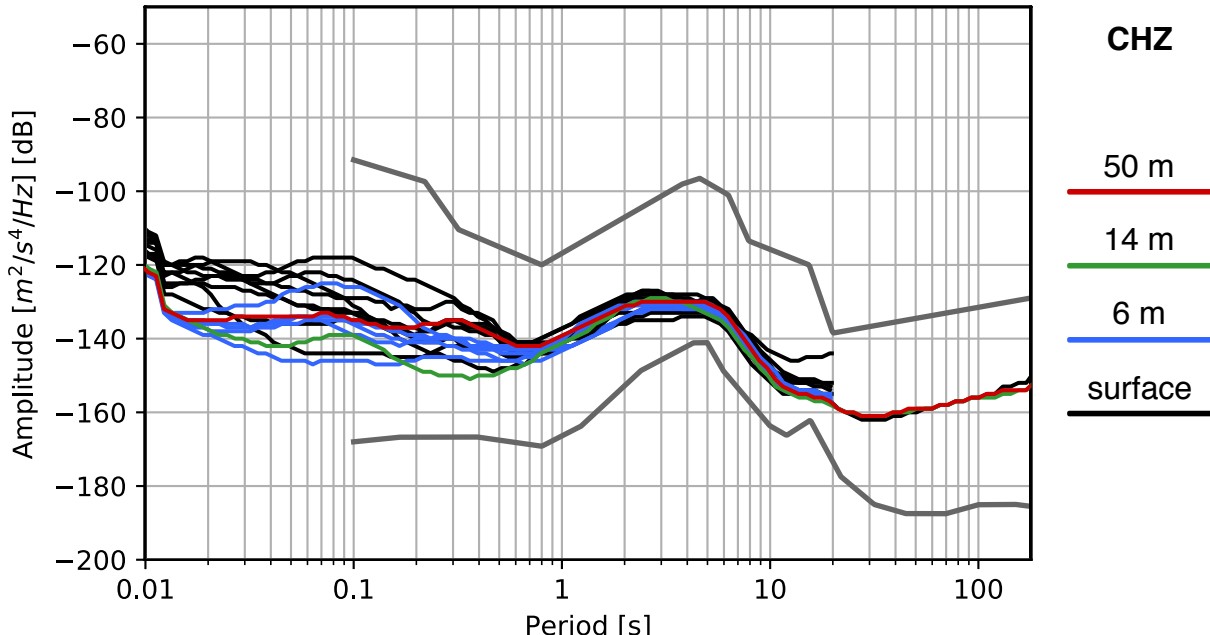

**Figure 10: Median values of PPSD computed for the vertical components (channel CHZ) of sensors installed at different depths and locations over the entire period of operation of the INSIEME seismic network. Black curves are the median values of PPSD computed when sensors were installed on surface; blue, green, and red curves are those referred to sensors installed at 6 m, 14 m, and 50 m depth, respectively. For sensors with flat response up to 20 s, the median curves are plotted up to that period. The two grey curves indicate the New High and Low Noise models.**

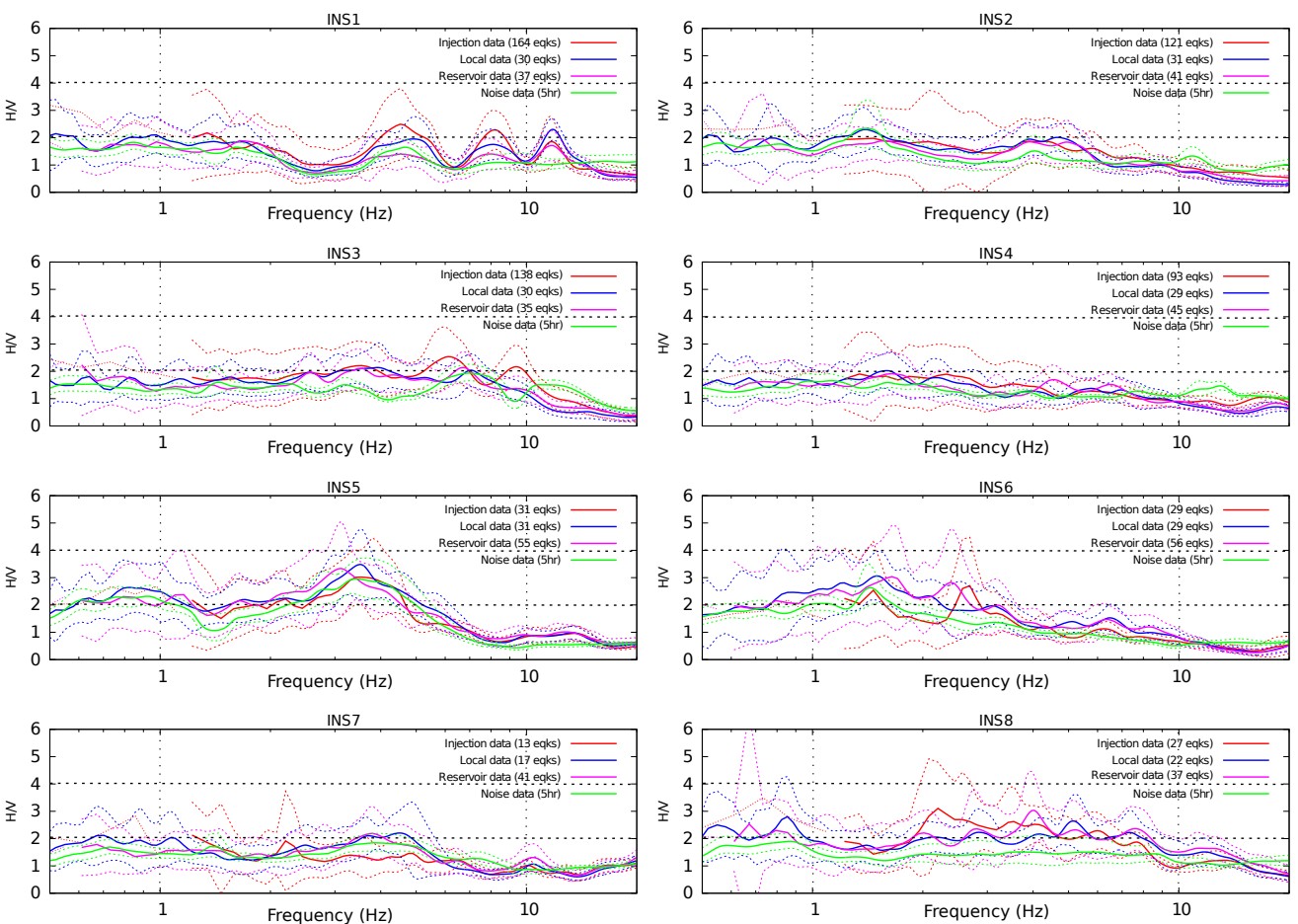

**Figure 11: HVSR and HVNSR curves computed at all the INSIEME network seismic stations. For each dataset, the number of used earthquakes and the hours of seismic noise are indicated. The solid coloured lines represent the average HVSR and HVNSR curves, whereas the dashed lines identify the ±1σ standard deviations.**

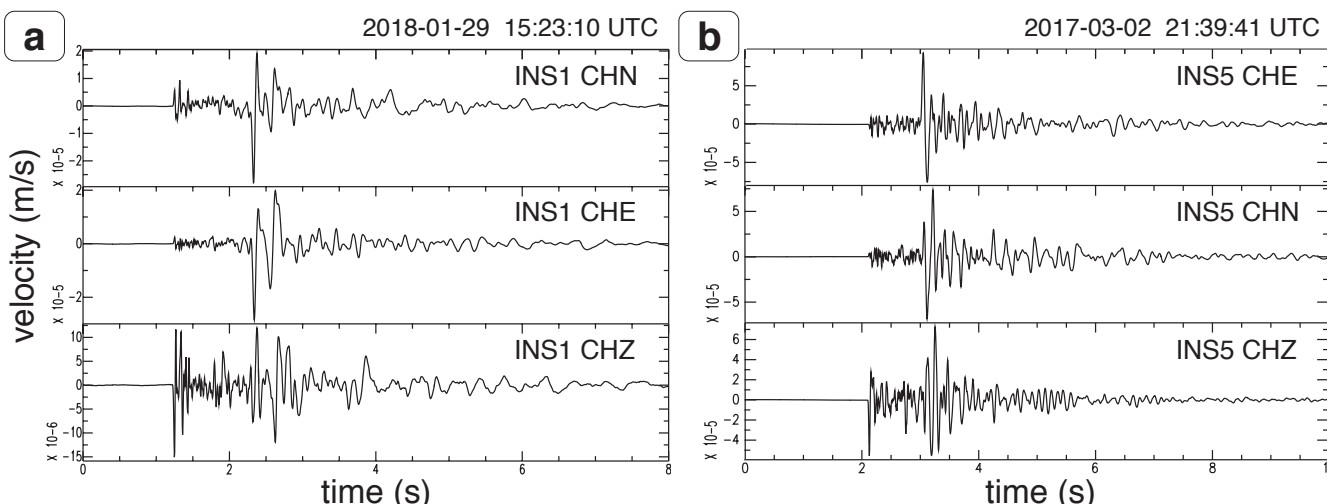

**Figure 12:** (a) The largest injection induced event (Ml=1.4, Lat=40.3182°N, Lon=15.9842°E, depth=3.50 km) recorded by the INS1 seismic station at 1.4 km epicentral distance, and (b) the largest reservoir induced event (Ml=1.8, Lat=40.2723°N, Lon=15.8840°E, depth=4.51 km) recorded by the INS5 seismic station at 1.9 km epicentral distance. On the top of the figures the seismic event origin time is reported. For station INS1 the original horizontal components were rotated counter-clockwise by an angle of 307.8° with respect to the North, according to the computations described in detail in section 2.2. A Ts – Tp of about 1 s can be clearly noticed for both the injection and the reservoir induced events at the correspondent closest station.

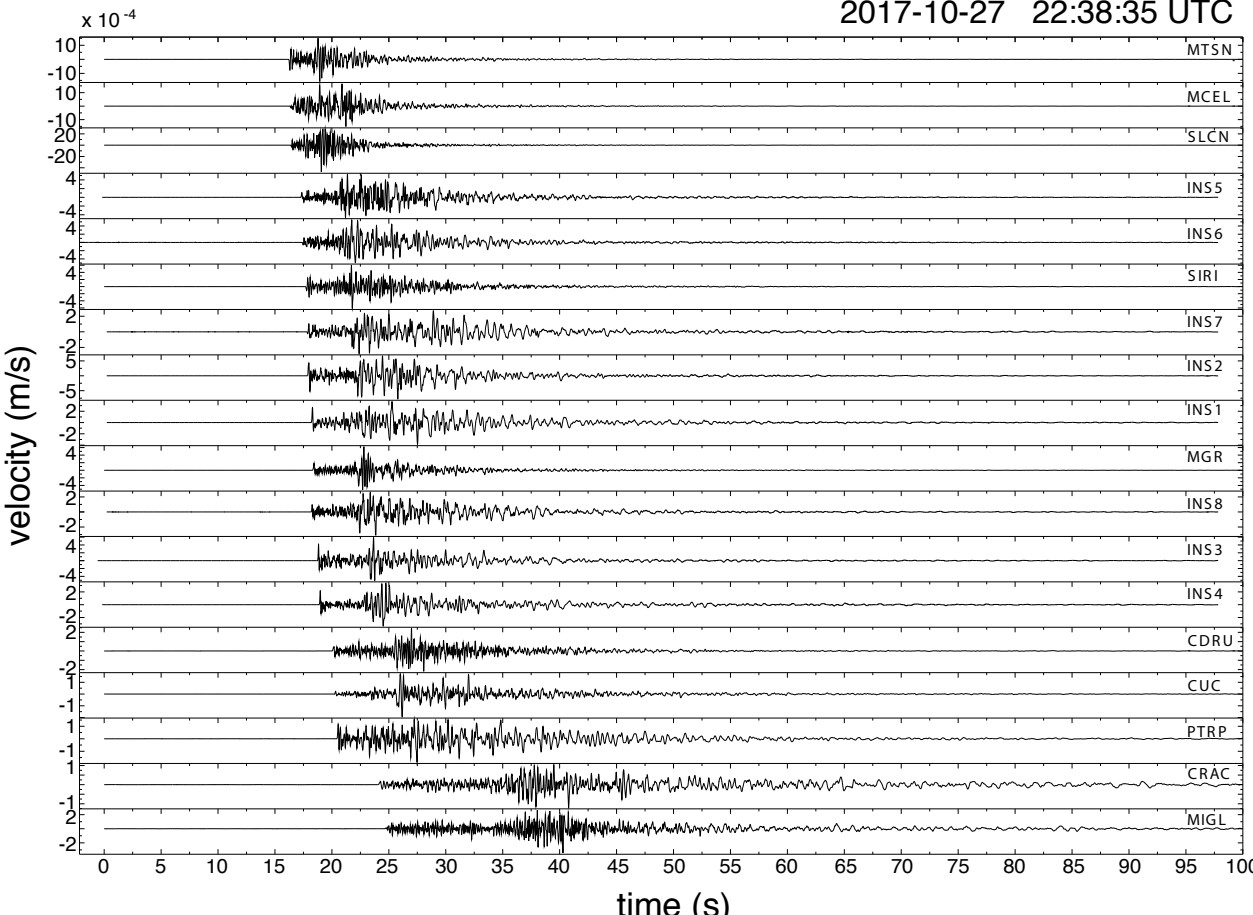

**Figure 13: Mw=3.8 local earthquake (Ml=4.0, Lat=40.3040˚N, Lon=15.7200˚E, depth=12.10 km) recorded by the vertical components of the stations belonging to the virtual seismic network composed of INSIEME, INGV and ISNet seismic stations. The traces are sorted, from top to the bottom, based on the first P-wave arrival time.**

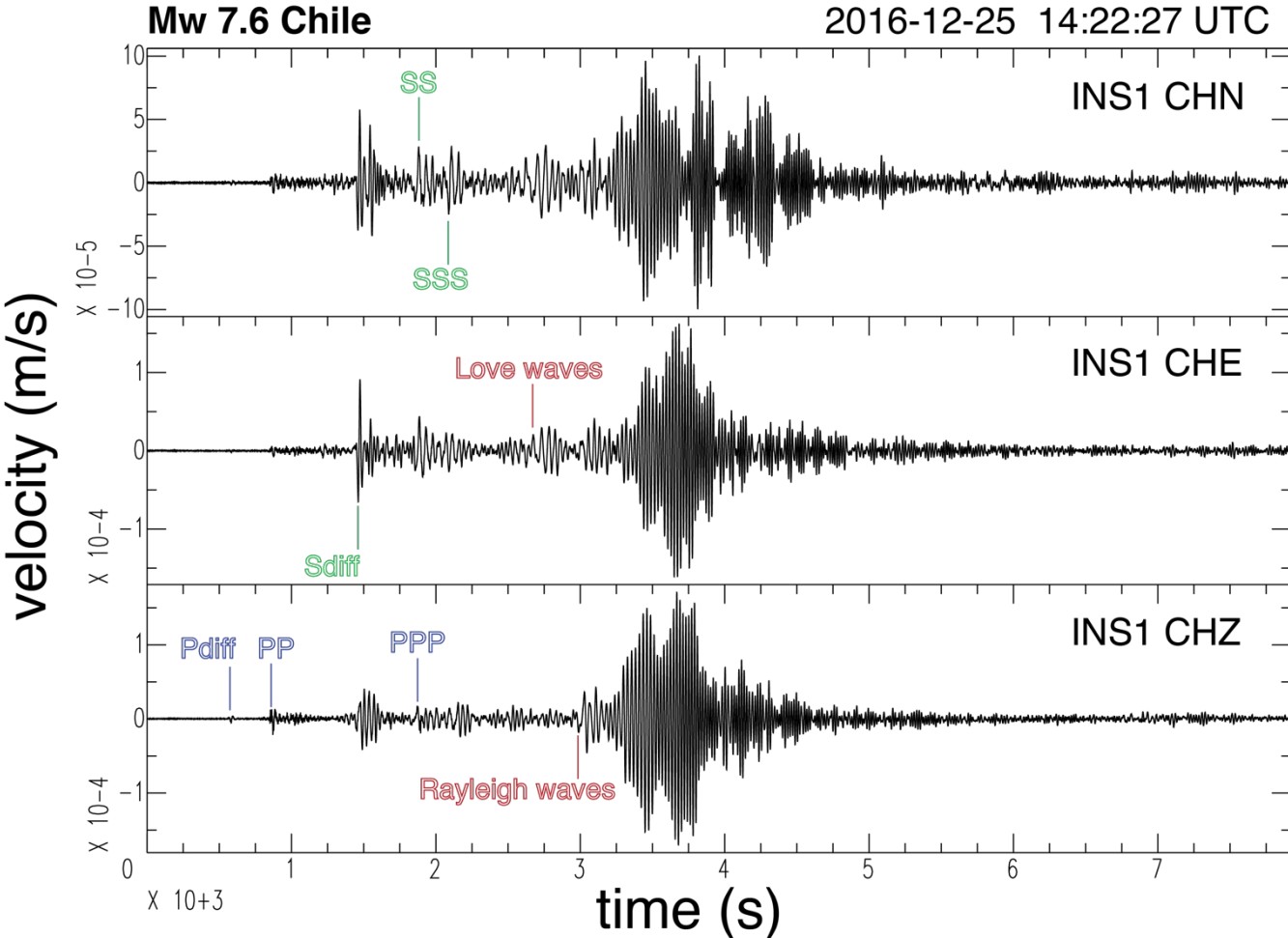

**Figure 14:** Mw=7.6 Chile earthquake of 2016-12-25 recorded by the three components of the INS1 seismic station. On the top right of the figure the origin time of the teleseism is reported. The theoretical arrival times of the most important recorded seismic phases are shown. The original horizontal components were rotated counter-clockwise by an angle of 307.8° with respect to the North, according to the computations described in detail in section 2.2.