# Peer review of "The INSIEME seismic network: a research infrastructure for studying induced seismicity in the High Agri Valley (southern Italy)"

_Earth System Science Data, 2019_

## Referee Comment (RC1) · Anonymous Referee #1 · 28 Jul 2019

The aim of this paper is to describe a local seismic network for the observation of induced seismicity at the High Agry Valley in S. Italy. The waveform data that is recorded from the associated stations is open and induced seismicity is an interesting topic with serious implications for the local communities. The paper covers topics such as the installation process and technical characteristics of the associated seismic stations, as well as the site effect characteristics. My main points are (i) the addition of a new figure showing an example of a station's response (amplitude/phase) even though the dataless station files are available in the supplementary material, and (ii) that the discrimination of induced events and local earthquakes is being done based on the hypocentre depth only. I think that the paper would benefit from a brief source mechanism study

for the discussed (or a sample) seismic events. I have made other minor comments (see below) which have to do mainly with the English language syntax throughout the manuscript. Overall the scientific content is good and useful and I recommend moderate/major revisions of the paper. I suggest the authors to proof read the manuscript very carefully upon submission of the revised manuscript.

p.2 - l.2-6 Induced seismicity is commonly accepted to be anthropogenic. I think McGarr(2002) discusses whether different cases are induced or not and in which degree, but in general he accepts induced seismicity as anthropogenic - please rephrase.

p.2 - l.19 and the discrimination between natural and induced seismicity. I think this should go to b) from a pure scientific point of view.. I don't think this has to do much with social and economic impacts.

p.3 -l.14 remove instead.

p.3 -l.20 with the highest seismogenic potential... I think this term is described better in terms of enercgy accumulation ..or motion rate mm/year? The expected maximum acceleration has to do also with local site conditions - maybe rephrase.

p.4 - l.10 regular azimuthal coverage and distribution as regular as possible.. I believe the authors mean uniform aziumthal distribution - please clarify and rephrase.

p.4 l.15 - 20 - the average distance between stations.. I think the same point is repeated twice here, please correct this. Moreover, it is better to give the depth range on the second point where you first discuss the importance of depth and epicentral distance (l.13).

p.5 - l.1 remove Afterwards

p.5 - l.3 ...as more constraints as possible... correct to ..as many constraints as possible

p.5 - Subsection 2.2 I think the first part of the first paragraph does not read very well in my opinion. Please replace by: Considering that the main target of the INSIEME

network is to detect and locate the anthropogenic microseismicity in the HAV (Ml ≤ 2.7), the seismic stations were equipped with triaxial weak-motion broadband sensors: six 0.05-100 Hz and two 0.0083-100 Hz Trillium Compact Posthole (TCPH) seismometers which provide a flat response to ground velocity up to 100 Hz. The data-loggers are Centaur Digital Recorders with a dynamic range of 140 dB. All seismometers and data-loggers are manufactured by Nanometrics Inc. (see Table 1). Continuous acquisition of digital waveforms is provided by the INSIEME network at 250 Hz sampling rate.

p.5 - l.15 Even though the Nyquist frequency is well beyond the instrument's flat response high end I am wondering what is the phase response especially in the high frequencies from the instruments' sensitivity frequency to 125 Hz. What is the target frequency range in this study? There are a few broadband instruments currently in the IRIS data services showing strange phase responses even close to 1Hz. I believe a figure showing the amplitude and phase response of one station would be a good addition.

p.5 - l.22 ...the Winter season (see Figure 2a), then the solar.. start new sentence: ...the winter season (see Figure 2a). The solar...

p.5 - l.30 what is this system? please give a brief description.

p.6 -l.1 is highly deviated over 20 m depth - not very clear, please rephrase

p.6 - l.3 ..seismometers model, which operates.. change to: ..seismometers which operate..

p.6 - l.15 remove Afterwards

p.6 -l.27 was only 70 m distance from the borehole sensor.. change to: was only 70 m away from the borehole sensor..

p.6 - l. 28 ..and acquired simultaneously with station INS1 from 2016-10-12 to 2017-01-24 I think the authors mean that these stations were in operation during the same period of time - please rephrase

p.6 -l.30 please provide a numerical description of you calculations

p.7 -l.4 see similar comment at p.6 - l.28

p.7 - l. 5-6 teleseisms - please change to: teleseismic events

p.7 -l.21 Nanometric Centaur digital recorder... correct to: The Nanometrics Centaur digital recorder..

p.7 - l.25 ..that prevent the internet connection - please rephrase

p.7 -l.26 disconnects for few seconds.. for a few seconds

p.8 - section 2.3 (last paragraph) use collect instead of gather Why some events cannot be located? please explain briefly

p. 9 l. 22 ..compared to each other

p.9 l. 22-23 ..the noise level is more regular at 50 m depth - what does regular noise level mean? please rephrase

p.10 l.8 ..when both the stations - remove "the", remove "respective"

p.10 l.32-33 ..In that way, we guaranteed... please rephrase

p.11 - l.1 why is date time seismic noise is being used? please explain

p.11 eq. 2 should be HVSR =

p.11 - l.28 ..that the most.. remove "the"

p.11 - l.32 did the authors calculate NS and EW HVSRs separately to investigate any directivity and azimuthal effects? If yes, were they found negligible?

p.12 -l.11 competent rocks - Is there a better term to describe this?

p.12 - l.18 ..located in the 1-D velocity model by Improta et al. (2017) by adopting Hypo71.. change to ..using the 1-D velocity model by Improta et al. (2017) and

Hypo71..

p.12 -l.20 To this purpose, and particularly to better locate.. change to In order to better locate local events outside the...

p.12 - l.20 what is the distance between stations of the virtual network? Maybe a new figure showing the distribution of the "virtual network" and the INSIEME network together could be shown at the supplementary material.

p.12 -l.33 ..related to an earthquake occurred on 2018-01-29.. I think the authors mean that this is an induced event. Maybe it would have been more appropriate not to use the term earthquake and simply refer to it as an induced seismic event, similar to the line above (l.32) ..from preliminary event location..

p.13 -l.11 similar as in my previous comment (replace earthquake with event)

p.13 - paragraph 2. The authors discriminate the induced events from local earthquakes using the depth as their main criterion. Did the authors attempt to determine the focal mechanisms of any of these events, by means of first motion polarities and/or amplitude ratios for example? Is there a high signal-to-noise ratio on the INSIEME stations and the virtual seismic network recordings to do so? please add an example, if not please justify your answer.

p.13 l-32 replace Dziewonsky with Dziewonski.

p.14 - l.5 ..we have decided to do not uninstall the network.. change to ..we have decided not to uninstall the network..

p.14 - l.11 the begin of data.. change to ..the beginning of data..

p.14 - l.11 very attractive area.. attractive in which manner? maybe change to very interesting area

p.14 - l.15 ..consisting in.. change to consisting of..

p.14 -l.16 and two 120s-100Hz Trillium Compact Posthole sensors,

p.14 l.18-19 ..started to have troubles.. please rephrase (e.g. presented an intermittent fault)

p.14 - l.24-25 ..with negligible site amplification..

Fig. 1 I am not sure if the last sentence in the caption is necessary, maybe move it to the Acknowledgements.

Fig.5 caption: Below each actual... I think the authors could rephrase the caption beyond that point. It is not very clear to me.

Fig.8 caption: The solid coloured lines...

Fig. 9 caption: replace earthquake with event

Fig. 10 caption: ..from top to the bottom,

As a general rule when the authors refer to the number of objects (e.g., stations) which is less than ten, please write this as a word. If this number is higher than ten, you can write it as a number.

---

## Author Comment (AC1) · 7 Aug 2019

The aim of this paper is to describe a local seismic network for the observation of induced seismicity at the High Agri Valley in S. Italy. The waveform data that is recorded from the associated stations is open and induced seismicity is an interesting topic with serious implications for the local communities. The paper covers topics such as the installation process and technical characteristics of the associated seismic stations, as well as the site effect characteristics. My main points are (i) the addition of a new figure showing an example of a station's response (amplitude/phase) even though the dataless station files are available in the supplementary material, and (ii) that the discrimination of induced events and local earthquakes is being done based on the hypocentre depth only. I think that the paper would benefit from a brief source mechanism study for the discussed (or a sample) seismic events. I have made other minor comments (see below) which have to do mainly with the English language syntax throughout the manuscript. Overall the scientific content is good and useful and I recommend moderate/major revisions of the paper. I suggest the authors to proof read the manuscript very carefully upon submission of the revised manuscript.

REPLY: We would like to thank you for your useful major and minor comments that surely improve the quality and the readability of the paper. Even if we do not completely agree with some few comments, they were important to better clarify some sentences or to provide additional information in the paper. Regarding the main point (i), as requested, we will add in the supplementary material of the revised manuscript a new figure showing a station's response both in amplitude and phase. Regarding the main point (ii) we would like to highlight that the classification of induced events is not based merely on their focal depth but also by evaluating if the event belongs to the cluster already classified as induced from previous studies. Probably this aspect was not evinced in the original manuscript, therefore we will add few sentences in the revised version to better describe how new recorded events are classified as induced. Furthermore, we will provide in the supplementary material some focal mechanisms computed with SeisComp3 for a sample of induced events (both fluid-injection and reservoir-induced events), and we will discuss the results in the text. Finally, we will carefully proof read the revised manuscript. Please, let us know if you need additional clarifications. Following we will provide a point-by-point reply to each minor comment. Each reply is preceded by the word "REPLY":

p.2 - l.2-6 Induced seismicity is commonly accepted to be anthropogenic. I think Mc-Garr(2002) discusses whether different cases are induced or not and in which degree, but in general he accepts induced seismicity as anthropogenic - please rephrase.

REPLY: Probably our sentence was too short and not clear enough. In the framework of "anthropogenic seismicity" McGarr (2002) provided a specific significance to the adjectives "induced", "triggered", and "stimulated": «As used here, the adjective "induced" describes seismicity resulting from an activity that causes a stress change that is comparable in magnitude to the ambient shear stress acting on a fault to cause slip, whereas "triggered" is used if the stress change is only a small fraction of the ambient level. By "stimulated" we refer generally to seismicity either triggered or induced by human activities». Therefore, using the word induced as synonymous with anthropogenic is in principle technically incorrect. Anyway, nowadays is commonly accepted to interchange the two words for the sake of simplicity. We will rephrase the sentence with the aim to better explain this concept that allows us to use "induced" as well as "anthropogenic" along the whole manuscript.

p.2 - l.19 and the discrimination between natural and induced seismicity. I think this should go to b) from a pure scientific point of view... I don't think this has to do much with social and economic impacts.

REPLY: We prefer to leave it in a) because in our opinion the discrimination between natural and induced seismicity has both strong social and economic implications. Indeed, establishing cause and effect between recorded earthquakes and local energy technologies is an important information for insurance and legal purposes to assess if a damaging earthquake is associated or not to the energy technology in operation nearby the damaged area. As an example, after the occurrence of the Ml=3.4 induced earthquake in Basilea, several damage claims arose from local residence and the company's liability insurance payed more than $9 million for damages (e.g., see Grigoli et al., 2017, and references therein). Furthermore, we have to bear in mind that people can tolerate the occurrence of non-damaging natural events but they do not tolerate induced felt events even if any damage does not occur.

p.3 -l.14 remove instead.

REPLY: Ok, thank you.

p.3 -l.20 with the highest seismogenic potential... I think this term is described better in terms of energy accumulation ...or motion rate mm/year? The expected maximum acceleration has to do also with local site conditions - maybe rephrase.

REPLY: The expected maximum acceleration referred to average hard ground conditions is the parameter used by the Italian national reference seismic hazard model (Gruppo di Lavoro MPS, 2004); therefore, we use the same parameter. We will clarify the definition by modifying the sentence from "...with an expected maximum acceleration for an exceedance probability of 10%..." to "...with an expected maximum acceleration (referred to average hard ground conditions) for an exceedance probability of 10%...". Furthermore, we will add at p.3 - l.25, the following sentence: "Furthermore, it has been estimated from GPS velocity and strain rate field data (D'Agostino, 2014) that the extensional opening in the axial part of southern Apennines is about 3 mm/yr.".

p.4 - l.10 regular azimuthal coverage and distribution as regular as possible... I believe the authors mean uniform azimuthal distribution - please clarify and rephrase.

REPLY: Thank you. We will substitute "with a regular azimuthal coverage and distribution as regular as possible" with "with as uniform as possible azimuthal distribution".

p.4 l.15 - 20 - the average distance between stations. I think the same point is repeated twice here, please correct this. Moreover, it is better to give the depth range on the second point where you first discuss the importance of depth and epicentral distance (l.13).

REPLY: The point at l.13 imposes the constraint on the minimum distance of the nearest station to the event whereas the point at l.15 is referred to the minimum distance among stations. Both the constraints are dependent from the event focal depth, so it is better to introduce them together. We will merge the two points (l.13 and l.15) and we clarify the sentence.

p.5 - l.1 remove Afterwards

REPLY: Ok, thank you.

p.5 - l.3 ...as more constraints as possible... correct to . . .as many constraints as possible

REPLY: Thank you. We accept your suggestion.

p.5 - Subsection 2.2 I think the first part of the first paragraph does not read very well in my opinion. Please replace by: Considering that the main target of the INSIEME network is to detect and locate the anthropogenic microseismicity in the HAV (Ml $\leq$ 2.7), the seismic stations were equipped with triaxial weak-motion broadband sensors: six 0.05-100 Hz and two 0.0083-100 Hz Trillium Compact Posthole (TCPH) seismometers which provide a flat response to ground velocity up to 100 Hz. The data-loggers are Centaur Digital Recorders with a dynamic range of 140 dB. All seismometers and data-loggers are manufactured by Nanometrics Inc. (see Table 1). Continuous acquisition of digital waveforms is provided by the INSIEME network at 250 Hz sampling rate.

REPLY: Thank you very much for your effort. We will replace the sentence according to your comment.

p.5 - l.15 Even though the Nyquist frequency is well beyond the instrument's flat response high end I am wondering what is the phase response especially in the high frequencies from the instruments' sensitivity frequency to 125 Hz. What is the target frequency range in this study? There are a few broadband instruments currently in the IRIS data services showing strange phase responses even close to 1Hz. I believe a figure showing the amplitude and phase response of one station would be a good addition.

REPLY: We will add in the supplements a figure showing the amplitude and phase response of one station. Regarding the target frequency range, our goal is to provide recordings with a range as broad as possible. The high frequency bound allows us to estimate the corner frequency of small events (if we are able to correctly remove

the attenuation); the low frequency bound allows the seismologists to use the data also for other applications (e.g., evaluation of possible site amplification effects at low frequencies, ambient seismic noise tomography, recordings of teleseismic events, etc.).

p.5 - l.22 ...the Winter season (see Figure 2a), then the solar.. start new sentence: ...the winter season (see Figure 2a). The solar...

REPLY: Thank you. We will fix it.

p.5 - l.30 what is this system? please give a brief description.

REPLY: Ok, we will provide a brief description of the system.

p.6 -l.1 is highly deviated over 20 m depth - not very clear, please rephrase

REPLY: Ok, we will rephrase the sentence. We will write that the inclination of the borehole becomes greater than the maximum operational tilt of the sensor.

p.6 - l.3 . . .seismometers model, which operates. . . change to: . . .seismometers which operate. . .

REPLY: Thank you. We will fix it.

p.6 - l.15 remove Afterwards

REPLY: Ok, thank you.

p.6 -l.27 was only 70 m distance from the borehole sensor. . . change to: was only 70 m away from the borehole sensor. . .

REPLY: Thank you. We accept your suggestion.

p.6 - l. 28 ...and acquired simultaneously with station INS1 from 2016-10-12 to 2017-01-24 I think the authors mean that these stations were in operation during the same period of time - please rephrase

REPLY: Yes, you are right. We will rephrase the sentence.

p.6 -l.30 please provide a numerical description of your calculations

REPLY: Ok, we will provide a numerical description of our calculation.

p.7 -l.4 see similar comment at p.6 - l.28

REPLY: Ok. We will rephrase the sentence.

p.7 - l. 5-6 teleseisms - please change to: teleseismic events

REPLY: The word "teleseism" is correct. Anyway, we will write "teleseismic event" along the whole text.

p.7 -l.21 Nanometric Centaur digital recorder... correct to: The Nanometrics Centaur digital recorder..

REPLY: Ok. Thank you.

p.7 - l.25 ..that prevent the internet connection - please rephrase

REPLY: We will substitute "problems that prevent the internet connection" with "problems causing the interruption of the internet connection"

p.7 -l.26 disconnects for few seconds.. for a few seconds

REPLY: Ok. Thank you.

p.8 - section 2.3 (last paragraph) use collect instead of gather Why some events cannot be located? please explain briefly

REPLY: We will substitute "gathered" with "collected". Some small events can be only detected because we do not have enough P- and S-wave pickings for locating them. As an example, if we have an event recorded by only one or two stations the event is not locatable.

p. 9 l. 22 ..compared to each other

REPLY: We will fix it.

p.9 l. 22-23 ..the noise level is more regular at 50 m depth - what does regular noise level mean? please rephrase

REPLY: Ok. We will rephrase the sentence. With "regular noise" we would like to explain that during time the noise does not change so much.

p.10 l.8 ..when both the stations - remove "the", remove "respective"

REPLY: We will fix it.

p.10 l.32-33 ..In that way, we guaranteed... please rephrase

REPLY: We will substitute the sentence "In that way, we guaranteed that the highest number of stations had recorded the selected data." with "Indeed, the greater the earthquake is the greater is the number of recording stations."

p.11 - l.1 why is date time seismic noise is being used? please explain

REPLY: We are not sure if we rightly understood this question. Do you mean "why did you specify the date of the seismic noise recordings"? If so, we decided to specify it in order to allow the reproducibility of the analyses. Otherwise, if you mean "day" instead of "date", we decided to show the results obtained by one of the ambient noise data streams that we selected for the analyses. Indeed, we carried out the analyses by using different noise data streams (night time data, day time data, very noisy day data, low noisy day data, etc.), obtaining consistent results. Attached Figure 1 shows an example of results related to INS1 station by using data characterized by low level of noise (left panel) and high level of noise (right panel). Black and red lines in Figure 1 indicate the results obtained using day time and night time noise data streams, respectively. The dashed lines identify the +- one standard deviation.

p.11 eq. 2 should be HVSR =

REPLY: Yes, thank you.

p.11 - l.28 ..that the most.. remove "the"

REPLY: Ok. We will remove "the".

p.11 - l.32 did the authors calculate NS and EW HVSRs separately to investigate any directivity and azimuthal effects? If yes, were they found negligible?

REPLY: We thank you for this comment. Yes, we already separately computed the HVSR taking into account only the NS and the EW component. As an example, we show in attached Figure 2 the comparison between the retrieved HVSR curves obtained using the NS and EW component, respectively, of reservoir data recordings at INS5 seismic station. Indeed, INS5 site is the one characterized by an amplification at about 3 Hz, as shown in figure 8 of the manuscript. As shown in Figure 2, we do not observe any azimuthal or directivity effects. Actually, the retrieved HVSR curves are very similar. As shown in the manuscript, the results do not depend from the earthquake category selection (IIE, RIE and LE).

p.12 -l.11 competent rocks - Is there a better term to describe this?

REPLY: We will substitute "at that depth a sharp lithological change between less and more competent rocks." with "at that depth a sharp lithological change from alluvial deposits to Gorgoglione Formation."

p.12 - l.18 ..located in the 1-D velocity model by Improta et al. (2017) by adopting Hypo71.. change to ..using the 1-D velocity model by Improta et al. (2017) and Hypo71..

REPLY: Thank you. We accept your suggestion.

p.12 -l.20 To this purpose, and particularly to better locate.. change to In order to better locate local events outside the...

REPLY: Thank you for your suggestion. We will fix it.

p.12 - l.20 what is the distance between stations of the virtual network? Maybe a new figure showing the distribution of the "virtual network" and the INSIEME network

together could be shown at the supplementary material.

REPLY: Thank you for your idea. We will update the supplemental file "INSIEME-network.kmz" by adding the locations of all the public and private seismic stations that can be used for the "virtual network". By clicking on each station one can interactively read additional details.

p.12 -l.33 ..related to an earthquake occurred on 2018-01-29.. I think the authors mean that this is an induced event. Maybe it would have been more appropriate not to use the term earthquake and simply refer to it as an induced seismic event, similar to the line above (l.32) ..from preliminary event location..

REPLY: We will fix it. Anyway, induced seismic events in the High Agri Valley are (micro)earthquakes.

p.13 -l.11 similar as in my previous comment (replace earthquake with event)

REPLY: We will fix it.

p.13 - paragraph 2. The authors discriminate the induced events from local earthquakes using the depth as their main criterion. Did the authors attempt to determine the focal mechanisms of any of these events, by means of first motion polarities and/or amplitude ratios for example? Is there a high signal-to-noise ratio on the INSIEME stations and the virtual seismic network recordings to do so? please add an example, if not please justify your answer.

REPLY: We provided a detailed answer to these questions in the first reply to your comments. We will add some examples of focal mechanisms in the supplementary material. Indeed, in some cases we have enough first motion polarities.

p.13 l-32 replace Dziewonsky with Dziewonski.

REPLY: Thank you for pointing out the typewriting error.

p.14 - l.5 ..we have decided to do not uninstall the network.. change to ..we have

decided not to uninstall the network..

REPLY: Thank you. We accept your suggestion.

p.14 - l.11 the begin of data.. change to ..the beginning of data..

REPLY: We will fix it.

p.14 - l.11 very attractive area.. attractive in which manner? maybe change to very interesting area

REPLY: You're right. We will fix it.

p.14 - l.15 ..consisting in.. change to consisting of..

REPLY: You're right. We will fix it.

p.14 -l.16 and two 120s-100Hz Trillium Compact Posthole sensors,

REPLY: We will fix it.

p.14 l.18-19 ..started to have troubles.. please rephrase (e.g. presented an intermittent fault)

REPLY: Ok. Thank you for your suggestion.

p.14 - l.24-25 ..with negligible site amplification..

REPLY: We will fix it.

Fig. 1 I am not sure if the last sentence in the caption is necessary, maybe move it to the Acknowledgements.

REPLY: This was required by the journal policy during the validation of the initial submission.

Fig.5 caption: Below each actual... I think the authors could rephrase the caption beyond that point. It is not very clear to me.

REPLY: The sentences "Below each actual PPSD there is visualized the data basis for the PPSD. The top row shows data fed into the PPSD: green patches represent available data, red patches (not in this case) represent eventual gaps in streams. The bottom row in blue shows the single PSD measurements that go into the histogram." are taken from the PPSD webpage describing how to read the figure (https://docs.obspy.org/tutorial/code_snippets/probabilistic_power_spectral_density.html). Anyway, we will simplify such description in the revised manuscript.

Fig.8 caption: The solid coloured lines...

REPLY: We will fix the error.

Fig. 9 caption: replace earthquake with event Fig. 10 caption: ..from top to the bottom,

REPLY: Ok. We will fix it.

As a general rule when the authors refer to the number of objects (e.g., stations) which is less than ten, please write this as a word. If this number is higher than ten, you can write it as a number.

REPLY: Thank you for your suggestion. We will write along the whole manuscript the number as a word if it is less than ten.

[Figure]

[Figure]

**Fig. 1.** HVNSR curves obtained for INS1 station by using data characterized by low level of noise (left panel) and high level of noise (right panel).

[Figure]

**Fig. 2.** Average HVSR curves retrieved by analyzing the recordings at station INS5 of the reservoir induced events: red and blue lines indicate the HVSR obtained for the NS and the EW component, respectively.

---

## Referee Comment (RC2) · Anonymous Referee #2 · 13 Dec 2019

General comments

Within this manuscript the authors describe a seismic network deployed in a region prone to induced seismicity, tailor made to better understand this process. Before describing the actual seismological infrastructure they provide an extensive introduction to the anthropogenic seismicity, to conclude with a short discussion about both scientific findings and a summary of the peculiarities of the collected dataset. Although being well organized the paper is still unbalanced towards scientific results rather than emphasizing the potential of this dataset for other users. The collected dataset made openly available to the community using standard formats and services has great value

and potential for the community to better understand the generation of induced seismicity and test alternative methods. The dataset should be the core of this manuscript without too many distractions for the readers about the own scientific findings of the authors. Considering the journal target and the high quality of the dataset described here I would suggest (in the detailed comments) a number of changes aiming at reducing the parts about the scientific findings while enhancing the presentation of this peculiar dataset.

Detailed comments

Page 1, line 27 "... Data collected until the end of the INSIEME project (2019-03-23) are already released ..." Indeed at IRIS DMC there are data to 25.06.2019. Check and correct if needed. What about real-time data? See also later comments about the Data availability section.

Page 1, line 29 "...available from (https://doi.org/10.7914/SN/3F_2016; Stabile et al., 2016)." Replace with "...available from IRIS DMC." Details about how to retrieve and use data should be provided with all details in the Data availability section.

Page 1 to page 2, line 9 Remove/reformulate the introduction aiming at keeping only three short paragraphs (~5 lines each) about: The project (Funder), the importance of high quality seismic networks to better understand induced seismicity and a short summary of the paper content (the actual lines 10-16 at page 2 can stay).

Figure 1 Change the color used for the INSIEME station to improve the visibility (currently with the dark blue is difficult to spot the triangles on the map).

Page 3, line 18 to page 4, line 5 This section, which is important to understand the context, could be included in the reshaped introduction.

Page 5, line 13 "They provide a flat response to ground velocity up to 100 Hz." Redundant, the higher limit is written already in the previous sentence.

Page 7, line 10-11 "Dataless of all the INSIEME seismic stations, which include comprehensive information of each station and the respective instrument response, are provided in the Supplement." This is not needed since metadata are provided in standard stationXML or text format. The authors can provide additional details (including URLs to station, dataselect and availability web services) in the data availability section.

Page 7, lines 15-20 The authors are providing here a long explanation about how to reach the remote with dynamic IPs. Do they try to use VPN? OpenVPN is supported by the hardware in use and they all connect to the same server. Add a sentence why they used this approach rather than creating a Virtual Private Network.

Page 7, line 23 "The router Teltonika RUT-500 is compliant with SeedLink, and therefore adopted as transmit tool" What's the meaning of this sentence? Probably that this hardware supports TCP/IP protocol?

Page 7, line 25 Probably the same can be achieved with the ping reboot functionality without the need to force daily reboots. Not sure though this version of router has this functionality.

Page 7, line 30 to page 8, line 7 Add a reference here to the data availability section where I suggest to add a figure with the data availability (%) for all stations for the entire period of operation of the network (e.g. using obspy-scan).

Page 8, line 27 Data quality Section and related figures 5 and 6 I suggest to add figures with Probability Density Functions for all stations/components and accordingly comment them in this section. The PDFs should be calculated over the entire period of operation of the network which according to the data available at IRIS DMC is 01.04.2016 - 25.06.2019. Current figures are only including 4 days of data which is not enough to have an idea about the actual data quality and argue about quality at different depths/locations. To facilitate the visualization of the figures median, 5th and 95th percentile should be plotted on each panel. To show the difference between the surface, shallow and deep installation (INS1 at 50 m) would be enough to have an additional figure with the comparison of the median values for sensors at different depths and locations.

Page 12, line 3 "...INS5 seismic station, where a small peak..." replace small with relevant. Indeed in figure 8 the H/V ratio exceeds 3, making this station the most amplified in the frequency band 2-5 Hz. Would have been interesting though to compute also spectral ratios among the stations having fixed one station as reference (e.g. INS4 or one station nearby free of site effects belonging the other permanent networks).

Page 14, line 1 Data availability section: this should evolve in a comprehensive description of the dataset availability. Provide details about where and how to access the data (fdsn web services at IRIS DMC). Data are available at IRIS DMC from 01.04.2016 to 25.06.2019. Is just this the open dataset described in this paper or this includes also open real-time data or periodic releases after certain embargo dates? Please specify in this section. A figure with the availability for the given period should be added. Would also be ideal to clearly state here if there is a license applied to the data and accordingly ask also IRIS DMC to include this in the DOI metadata of the network (including additional metadata as Funder, Sponsors, ORCIDs of the creators etc.). Moreover within the paper a seismic catalogue is mentioned and would be ideal to provide a link to it from here, either to an fdsn-event service or simply add catalogue to the supplementary material.

Page 14, lines 15 to page 1, line 3 Check this part carefully as most of this is redundant from the previous sections. Try to reduce redundancy and emphasize the part starting at page 15, line 4 stressing the peculiarities of this dataset.

Page 14, line 26 "... detected 856 local natural and induced earthquakes ..." Can the earthquake catalogue, obtained from this network, be added to the supplementary material? Alternatively can the authors point to a repository where this catalogue is hosted?

Supplementary material

Being the dataset archived in a FDSN data centre providing standard data and metadata formats datalessSEED volumes can be omitted in my opinion. Within the manuscript the authors are referring to an own earthquake catalogue. This would be indeed a useful addition for the supplementary material. Or at least a link to an open standard fdsn web service where this can be obtained.

---

## Author Response (AR1)

**Reply to Anonymous Referee #1 (PAPER: The INSIEME seismic network: a research infrastructure for studying induced seismicity in the High Agri Valley (southern Italy)" by Tony Alfredo Stabile et al.)**

*2020-01-17*

**Reply to General comments**

The aim of this paper is to describe a local seismic network for the observation of induced seismicity at the High Agri Valley in S. Italy. The waveform data that is recorded from the associated stations is open and induced seismicity is an interesting topic with serious implications for the local communities. The paper covers topics such as the installation process and technical characteristics of the associated seismic stations, as well as the site effect characteristics. My main points are (i) the addition of a new figure showing an example of a station's response (amplitude/phase) even though the data-less station files are available in the supplementary material, and (ii) that the discrimination of induced events and local earthquakes is being done based on the hypocentre depth only. I think that the paper would benefit from a brief source mechanism study for the discussed (or a sample) seismic events. I have made other minor comments (see below) which have to do mainly with the English language syntax throughout the manuscript. Overall the scientific content is good and useful and I recommend moderate/major revisions of the paper. I suggest the authors to proof read the manuscript very carefully upon submission of the revised manuscript.

REPLY: We would like to thank you for your useful comments that surely improve the quality and the readability of the paper. They were also important to better clarify some sentences or to provide additional information in the paper. Regarding the main point (i), as requested, we added in the revised manuscript a new figure (current Figure 2) showing a station's response both in amplitude and phase for the two type of broadband sensors. Regarding the main point (ii) we would like to highlight that the classification of induced events is not based merely on their focal depth but by evaluating if the event belongs to the cluster already classified as induced from previous studies. This aspect was not sufficiently well explained in the original manuscript, therefore we clarified it in the revised version. Furthermore, according to the General comments provided by Referee #2, the paper was unbalanced towards scientific results rather than on the potential of the dataset. Therefore, we added a sentence in section 5 emphasizing the potential use of this dataset for future source mechanism studies. Anyway, we provide in this reply some preliminary focal mechanisms (manually adjusted based on first arrival onset, using scolv tool of SeisComp3) for four natural and induced events (**Figures R3-R6**). Obviously, a comprehensive scientific study should be carried out in the future. Finally, we will carefully proofread the revised manuscript (we also updated Figures 8 and 10, current Figures 11 and 13, by correcting some label misalignments).

**Following we will provide a point-by-point reply to each minor comment. Each reply is colored in blue and is preceded by the word "REPLY". Figures attached to this reply are numbered from Figure R1 to Figure R6.**

**Reply to Detailed comments**

p.2 - l.2-6 Induced seismicity is commonly accepted to be anthropogenic. I think McGarr(2002) discusses whether different cases are induced or not and in which degree, but in general he accepts induced seismicity as anthropogenic - please rephrase.

REPLY: Probably our sentence was too short and not clear enough. In the framework of "anthropogenic seismicity" McGarr (2002) provided a specific significance to the adjectives "induced", "triggered", and "stimulated": <<As used here, the adjective "induced" describes seismicity resulting from an activity that causes a stress change that is comparable in magnitude to the ambient shear stress acting on a fault to cause slip, whereas "triggered" is used if the stress change is only a small fraction of the ambient level. By "stimulated" we refer generally to seismicity either triggered or induced by human activities>>. Therefore, using the word induced as synonymous with anthropogenic is in principle technically incorrect. Anyway, nowadays it is commonly accepted to interchange the two words for the sake of simplicity. We rephrased the sentence by avoiding such discussion which is not important in the framework of this paper.

p.2 - l.19 and the discrimination between natural and induced seismicity. I think this should go to b) from a pure scientific point of view... I don't think this has to do much with social and economic impacts.

REPLY: We prefer to leave it in a) because in our opinion the discrimination between natural and induced seismicity has both strong social and economic implications. Indeed, establishing cause and effect between recorded earthquakes and local energy technologies is an important information for insurance and legal purposes to assess if a damaging earthquake is associated or not to the energy technology in operation nearby the damaged area. As an example, after the occurrence of the Ml=3.4 induced earthquake in Basel, several damage claims arose from local residents and the company's liability insurance payed more than $9 million for damages (e.g., see Grigoli et al., 2017, and references therein). Furthermore, we have to bear in mind that people can tolerate the occurrence of non-damaging natural events but they do not tolerate induced felt events even if any damage does not occur.

p.3 -l.14 remove instead.

REPLY: Ok, thank you.

p.3 -l.20 with the highest seismogenic potential... I think this term is described better in terms of energy accumulation …or motion rate mm/year? The expected maximum acceleration has to do also with local site conditions - maybe rephrase.

REPLY: The expected maximum acceleration referred to average hard ground conditions is the parameter used by the Italian national reference seismic hazard model (Gruppo di Lavoro MPS, 2004); therefore, we use the same parameter. We clarified the definition by modifying the sentence from "…with an expected maximum acceleration for an exceedance probability of 10%..." to "…with an expected maximum acceleration (referred to average hard ground conditions) for an exceedance probability of 10%...". We also added the following sentence: "Furthermore, it has been estimated from GPS velocity and strain rate field data (D'Agostino, 2014) that the extensional opening in the axial part of southern Apennines is about 3 mm/yr.". All these sentences were moved in the introduction, according to the fifth detailed comment of Referee #2.

p.4 - l.10 regular azimuthal coverage and distribution as regular as possible… I believe the authors mean uniform azimuthal distribution - please clarify and rephrase.

REPLY: Thank you. We substituted "with a regular azimuthal coverage and distribution as regular as possible" with "with as uniform as possible azimuthal distribution".

p.4 l.15 - 20 - the average distance between stations. I think the same point is repeated twice here, please correct this. Moreover, it is better to give the depth range on the second point where you first discuss the importance of depth and epicentral distance (l.13).

REPLY: The point at l.13 imposes the constraint on the minimum distance of the nearest station to the event whereas the point at l.15 is referred to the minimum distance among stations. Both the constraints are dependent from the event focal depth, so it is better to introduce them together. We merged the two points (l.13 and l.15) and we clarified the sentence.

p.5 - l.1 remove Afterwards

REPLY: Ok, thank you.

p.5 - l.3 ...as more constraints as possible... correct to …as many constraints as possible

REPLY: Thank you. We accepted your suggestion.

p.5 - Subsection 2.2 I think the first part of the first paragraph does not read very well in my opinion. Please replace by: Considering that the main target of the INSIEME network is to detect and locate the anthropogenic microseismicity in the HAV (Ml ≤ 2.7), the seismic stations were equipped with triaxial weak-motion broadband sensors: six 0.05-100 Hz and two 0.0083-100 Hz Trillium Compact Posthole (TCPH) seismometers which provide a flat response to ground velocity up to 100 Hz. The data-loggers are Centaur Digital Recorders with a dynamic range of 140 dB. All seismometers and data-loggers are manufactured by Nanometrics Inc. (see Table 1). Continuous acquisition of digital waveforms is provided by the INSIEME network at 250 Hz sampling rate.

REPLY: Thank you very much for your effort. We replaced the sentence according to your comment.

p.5 - l.15 Even though the Nyquist frequency is well beyond the instrument's flat response high end I am wondering what is the phase response especially in the high frequencies from the instruments' sensitivity frequency to 125 Hz. What is the target frequency range in this study? There are a few broadband instruments currently in the IRIS data services showing strange phase responses even close to 1Hz. I believe a figure showing the amplitude and phase response of one station would be a good addition.

REPLY: We added in the revised version of the manuscript the new Figure 2 showing the amplitude and phase response of INS1 and INSX stations. Regarding the target frequency range, our goal is to provide recordings with a range as broad as possible. The high frequency bound allows us to estimate the corner frequency of small events (if we are able to correctly remove the attenuation); the low frequency bound allows the seismologists to use the data also for other applications (e.g., evaluation of possible site amplification effects at low frequencies, ambient seismic noise tomography, recordings of teleseismic events, etc.).

p.5 - l.22 ...the Winter season (see Figure 2a), then the solar.. start new sentence: ...the winter season (see Figure 2a). The solar...

REPLY: Thank you. We fixed it.

p.5 - l.30 what is this system? please give a brief description.

REPLY: Ok, we provided a brief description of the system.

p.6 -l.1 is highly deviated over 20 m depth - not very clear, please rephrase

REPLY: Ok, we rephrased the sentence.

p.6 - l.3 …seismometers model, which operates… change to: …seismometers which operate…

REPLY: Thank you. We fixed it.

p.6 - l.15 remove Afterwards

REPLY: Ok, thank you.

p.6 -l.27 was only 70 m distance from the borehole sensor… change to: was only 70 m away from the borehole sensor…

REPLY: Thank you. We accepted your suggestion.

p.6 - l. 28 ...and acquired simultaneously with station INS1 from 2016-10-12 to 2017- 01-24 I think the authors mean that these stations were in operation during the same period of time - please rephrase

REPLY: Yes, you are right. We rephrased the sentence.

p.6 -l.30 please provide a numerical description of your calculations

REPLY: Ok, we provided a numerical description of our calculation. We retrieved the average velocity of S-waves ($Vs=510$ m s$^{-1}$) for the first 50 m depth of the site hosting INS1 and INSX stations from Giocoli et al. (2015), and we compute the distance between the two sensors of station INS1 and INSX (d=88 m). Therefore, the reader is able to verify that $f_c$=0.5 Hz << V/d = 510 m s$^{-1}$ / 88 m = 5.8 Hz.

p.7 -l.4 see similar comment at p.6 - l.28

REPLY: Ok. We rephrased the sentence.

p.7 - l. 5-6 teleseisms - please change to: teleseismic events

REPLY: The word "teleseism" is correct. Anyway, we will write "teleseismic event" all along the text.

p.7 -l.21 Nanometric Centaur digital recorder... correct to: The Nanometrics Centaur digital recorder..

REPLY: Ok. Thank you.

p.7 - l.25 ..that prevent the internet connection - please rephrase

REPLY: We substituted "problems that prevent the internet connection" with "problems causing the interruption of the internet connection"

p.7 -l.26 disconnects for few seconds.. for a few seconds

REPLY: Ok. Thank you.

p.8 - section 2.3 (last paragraph) use collect instead of gather Why some events cannot be located? please explain briefly

REPLY: We substituted "gathered" with "collected". Some small events can be only detected because we do not have enough P- and S-wave pickings for locating them. As an example, if we have an event recorded by only one or two stations the event is not locatable. We added at the end of the sentence "based on the availability of both P- and S-wave pickings."

p. 9 l. 22 ..compared to each other

REPLY: We fixed it.

p.9 l. 22-23 ..the noise level is more regular at 50 m depth - what does regular noise level mean? please rephrase

REPLY: You're right. The sentence was not clear and we rephrased it. With "regular noise" we would like to explain that during time the noise does not change so much (less widespread).

p.10 l.8 ..when both the stations - remove "the", remove "respective"

REPLY: We fixed it.

p.10 l.32-33 ..In that way, we guaranteed... please rephrase

REPLY: We decided to remove the sentence "In that way, we guaranteed that the highest number of stations had recorded the selected data." Indeed, with this sentence we wanted to point out that local events with magnitude larger than 1.5 are recorded by all the stations.

p.11 - l.1 why is date time seismic noise is being used? please explain

REPLY: We are not sure if we correctly understood this question. Do you mean "why did you specify the date of the seismic noise recordings"? If so, we decided to specify it in order to allow the reproducibility of the analysis. Otherwise, if you mean "day" instead of "date", we decided to show the results obtained by one of the ambient noise data streams that we selected for the analysis. Indeed, we carried out the analysis by using different noise data streams (night time data, day time data, very noisy day data, low noisy day data, etc.), obtaining consistent results. Attached **Figure R1** shows an example of results related to INS1 station by using data characterized by low level of noise (left panel) and high level of noise (right panel). Black and

red lines in **Figure R1** indicate the results obtained using daytime and night time noise data streams, respectively. The dashed lines identify the +- one standard deviation.

p.11 eq. 2 should be HVSR =

REPLY: Yes, thank you.

p.11 - l.28 ..that the most.. remove "the"

REPLY: Ok. We removed "the".

p.11 - l.32 did the authors calculate NS and EW HVSRs separately to investigate any directivity and azimuthal effects? If yes, were they found negligible?

REPLY: We thank you for this comment. Yes, we already separately computed the HVSR taking into account only the NS and the EW component. As an example, we show in attached **Figure R2** the comparison between the retrieved HVSR curves obtained using the NS and EW component, respectively, of reservoir data recordings at INS5 seismic station. Indeed, INS5 site is the one characterized by an amplification at about 3 Hz, as shown in old Figure 8 (now Figure 11) of the manuscript. As shown in **Figure R2**, we do not observe any azimuthal or directivity effects. Actually, the retrieved HVSR curves are very similar. As shown in the manuscript, the results do not depend from the earthquake category selection (IIE, RIE and LE).

p.12 -l.11 competent rocks - Is there a better term to describe this?

REPLY: We will substitute "at that depth a sharp lithological change between less and more competent rocks." with "at that depth a sharp lithological change from alluvial deposits to Gorgoglione Formation."

p.12 - l.18 ..located in the 1-D velocity model by Improta et al. (2017) by adopting Hypo71.. change to ..using the 1-D velocity model by Improta et al. (2017) and Hypo71..

REPLY: Thank you. We accepted your suggestion.

p.12 -l.20 To this purpose, and particularly to better locate.. change to In order to better locate local events outside the...

REPLY: Thank you for your suggestion. We fixed it.

p.12 - l.20 what is the distance between stations of the virtual network? Maybe a new figure showing the distribution of the "virtual network" and the INSIEME network together could be shown at the supplementary material.

REPLY: Thank you for your idea. We updated the supplemental file "INSIEME-network.kmz" by adding the locations of all the public and private seismic stations that can be used for the "virtual network". By clicking on each station one can interactively read additional details. Furthermore, we added a map scale in Figure 1.

p.12 -l.33 ..related to an earthquake occurred on 2018-01-29.. I think the authors mean that this is an induced event. Maybe it would have been more appropriate not to use the term earthquake and simply refer to it as an induced seismic event, similar to the line above (l.32) ..from preliminary event location..

REPLY: We fixed it. Anyway, induced seismic events in the High Agri Valley are (micro)earthquakes.

p.13 -l.11 similar as in my previous comment (replace earthquake with event)

REPLY: We fixed it.

p.13 - paragraph 2. The authors discriminate the induced events from local earthquakes using the depth as their main criterion. Did the authors attempt to determine the focal mechanisms of any of these events, by means of first motion polarities and/or amplitude ratios for example? Is there a high signal-to-noise ratio on the INSIEME stations and the virtual seismic network recordings to do so? please add an example, if not please justify your answer.

REPLY: We provided a detailed answer to these questions in the first reply to your comments and we rephrased the sentence in the revised manuscript. We think that it is important to carry out a comprehensive study of source mechanisms, but this needs first an accurate relocation of seismicity and then a dedicated study. In some cases we have already enough first motion polarities to preliminary evaluate focal mechanisms (e.g., see **Figures R3-R6**) but, as soon as we will have also the data from the private seismic network, we will surely increase the number of computable focal mechanisms as well as their reliability.

p.13 l-32 replace Dziewonsky with Dziewonski.

REPLY: Thank you for pointing out the typewriting error.

p.14 - l.5 ..we have decided to do not uninstall the network.. change to ..we have decided not to uninstall the network..

REPLY: Thank you for your suggestion. Anyway, we have completely changed this sentence in the revised manuscript.

p.14 - l.11 the begin of data.. change to ..the beginning of data..

REPLY: We fixed it.

p.14 - l.11 very attractive area.. attractive in which manner? maybe change to very interesting area

REPLY: You're right. We fixed it.

p.14 - l.15 ..consisting in.. change to consisting of..

REPLY: You're right. We fixed it.

p.14 -l.16 and two 120s-100Hz Trillium Compact Posthole sensors,

REPLY: We fixed it.

p.14 l.18-19 ..started to have troubles.. please rephrase (e.g. presented an intermittent fault)

REPLY: Ok. Thank you for your suggestion. We rephrased the sentence.

p.14 - l.24-25 ..with negligible site amplification..

REPLY: We fixed it.

Fig. 1 I am not sure if the last sentence in the caption is necessary, maybe move it to the Acknowledgements.

REPLY: This was required by the journal policy during the validation of the initial submission.

Fig.5 caption: Below each actual... I think the authors could rephrase the caption beyond that point. It is not very clear to me.

REPLY: The sentences "Below each actual PPSD there is visualized the data basis for the PPSD. The top row shows data fed into the PPSD: green patches represent available data, red patches (not in this case)

represent eventual gaps in streams. The bottom row in blue shows the single PSD measurements that go into the histogram." are taken from the PPSD webpage describing how to read the figure (https://docs.obspy.org/tutorial/code_snippets/probabilistic_power_spectral_density.html). Anyway, we simplified such description in the revised manuscript.

Fig.8 caption: The solid coloured lines...

REPLY: We fixed the error.

Fig. 9 caption: replace earthquake with event Fig. 10 caption: ..from top to the bottom,

REPLY: Ok. We fixed it.

As a general rule when the authors refer to the number of objects (e.g., stations) which is less than ten, please write this as a word. If this number is higher than ten, you can write it as a number.

REPLY: Thank you for your suggestion. We wrote along the whole manuscript the number as a word when it was less than ten.

**FIGURES:**

[Figure]

*Figure R1*: HVNSR curves obtained for INS1 station by using data characterized by low level of noise (left panel) and high level of noise (right panel).

[Figure]

*Figure R2: Average HVSR curves retrieved by analyzing the recordings at station INS5 of the reservoir induced events: red and blue lines indicate the HVSR obtained for the NS and the EW component, respectively.*

[Figure]

**Figure R3:** *Preliminary focal mechanism of the Ml=1.4 fluid-injection induced event occurred on 2018-01-29 at 15:23:10 UTC.*

[Figure]

**Figure R4**: *Preliminary focal mechanism of the Ml=1.8 reservoir induced event occurred on 2017-03-02 at 21:39:41 UTC.*

[Figure]

**Figure R5**: *Preliminary focal mechanism of the Ml=1.6 local earthquake occurred on 2018-01-24 at 13:22:20 UTC.*

[Figure]

**Figure R6**: *Preliminary focal mechanism of the Ml=4.0 (Mw=3.8) local earthquake occurred on 2017-10-26 at 22:38:35 UTC.*

**Reply to Anonymous Referee #2 (PAPER: The INSIEME seismic network: a research infrastructure for studying induced seismicity in the High Agri Valley (southern Italy)" by Tony Alfredo Stabile et al.)**

*2020-01-17*

**Reply to General comments**

Within this manuscript the authors describe a seismic network deployed in a region prone to induced seismicity, tailor made to better understand this process. Before describing the actual seismological infrastructure they provide an extensive introduction to the anthropogenic seismicity, to conclude with a short discussion about both scientific findings and a summary of the peculiarities of the collected dataset. Although being well organized the paper is still unbalanced towards scientific results rather than emphasizing the potential of this dataset for other users. The collected dataset made openly available to the community using standard formats and services has great value and potential for the community to better understand the generation of induced seismicity and test alternative methods. The dataset should be the core of this manuscript without too many distractions for the readers about the own scientific findings of the authors. Considering the journal target and the high quality of the dataset described here I would suggest (in the detailed comments) a number of changes aiming at reducing the parts about the scientific findings while enhancing the presentation of this peculiar dataset.

REPLY: We would like to thank you for your useful comments that surely improve the quality and the readability of the paper. They were also important to better clarify some sentences, to provide additional information in the paper and in the metadata, and to better focus the paper on data instead of scientific findings. We carefully proofread the text (we also updated Figures 8 and 10, current Figures 11 and 13, by correcting some label misalignments).

**Following we will provide a point-by-point reply to each detailed comment. Each reply is colored in blue and is preceded by the word "REPLY".**

**Reply to Detailed comments**

Page 1, line 27 ". . . Data collected until the end of the INSIEME project (2019-03-23) are already released . . ." Indeed at IRIS DMC there are data to 25.06.2019. Check and correct if needed. What about real-time data? See also later comments about the Data availability section.

REPLY: Data after 2019-03-23 are partially already uploaded to IRIS DMC, but they will be shortly upgraded with a new station name and with the new VD network code. In the next future data of the permanent network VD will be transmitted in real time. We mentioned the VD permanent network in the Abstract, in the Data availability section, and in Discussion and conclusions section.

Page 1, line 29 ". . .available from (https://doi.org/10.7914/SN/3F_2016; Stabile et al., 2016)." Replace with ". . .available from IRIS DMC." Details about how to retrieve and use data should be provided with all details in the Data availability section.

REPLY: We partially modified the sentence according to your suggestion. We left the text in parenthesis because it is mandatory (required by the journal).

Page 1 to page 2, line 9 Remove/reformulate the introduction aiming at keeping only three short paragraphs (∼5 lines each) about: The project (Funder), the importance of high quality seismic networks to better understand induced seismicity and a short summary of the paper content (the actual lines 10-16 at page 2 can stay).

REPLY: We reformulate the introduction by considering also your detailed comment #5.

Figure 1 Change the color used for the INSIEME station to improve the visibility (currently with the dark blue is difficult to spot the triangles on the map).

REPLY: Thank you for your useful suggestion. We modified Figure 1 accordingly. We also included a map scale in the figure.

Page 3, line 18 to page 4, line 5 This section, which is important to understand the context, could be included in the reshaped introduction.

REPLY: Done. See also your detailed comment #3.

Page 5, line 13 "They provide a flat response to ground velocity up to 100 Hz." Redundant, the higher limit is written already in the previous sentence.

REPLY: We rephrased the sentence according to the suggested version provided by Referee #1.

Page 7, line 10-11 "Dataless of all the INSIEME seismic stations, which include comprehensive information of each station and the respective instrument response, are provided in the Supplement." This is not needed since metadata are provided in standard stationXML or text format. The authors can provide additional details (including URLs to station, dataselect and availability web services) in the data availability section.

REPLY: We removed the sentence and the datalessSEED volumes from the Supplement. We modified Data availability section according to your comments.

Page 7, lines 15-20 The authors are providing here a long explanation about how to reach the remote with dynamic IPs. Do they try to use VPN? OpenVPN is supported by the hardware in use and they all connect to the same server. Add a sentence why they used this approach rather than creating a Virtual Private Network.

REPLY: We explained in the revised manuscript the motivation of our technical choice. Furthermore, we have already some routers in our warehouse that can temporally replace a Teltonika in case of failure, but they do not support VPN.

Page 7, line 23 "The router Teltonika RUT-500 is compliant with SeedLink, and therefore adopted as transmit tool" What's the meaning of this sentence? Probably that this hardware supports TCP/IP protocol?

REPLY: Yes, Teltonika RUT-500 support TCP/IP protocol as all the routers. We removed the sentence and modified the previous one.

Page 7, line 25 Probably the same can be achieved with the ping reboot functionality without the need to force daily reboots. Not sure though this version of router has this functionality.

REPLY: Yes, we forgot to describe also this one. Actually, we have two automatic reboot systems. The first one is integrated inside the router and based on a ping utility called "watch dog timer": if the system does not ping an external public IP for some time, the router is automatically rebooted. The second one is based on an external programmable time switch that periodically (in our case once a week) unplugs for a few seconds the power supply of the router, thus preventing any software bug that could freeze the Teltonika. We added this description in the revised version of the manuscript.

Page 7, line 30 to page 8, line 7 Add a reference here to the data availability section where I suggest to add a figure with the data availability (%) for all stations for the entire period of operation of the network (e.g. using obspy-scan).

REPLY: Thank you for your suggestion. We added a figure with the data availability (Figure 4 of the revised manuscript) and use this figure to better describe data gaps in this section. We added a reference to Figure 4 also in Data availability section.

Page 8, line 27 Data quality Section and related figures 5 and 6 I suggest to add figures with Probability Density Functions for all stations/components and accordingly comment them in this section. The PDFs should be calculated over the entire period of operation of the network which according to the data available at IRIS DMC is 01.04.2016 - 25.06.2019. Current figures are only including 4 days of data which is not enough to have an idea about the actual data quality and argue about quality at different depths/locations.

To facilitate the visualization of the figures median, 5th and 95th percentile should be plotted on each panel. To show the difference between the surface, shallow and deep installation (INS1 at 50 m) would be enough to have an additional figure with the comparison of the median values for sensors at different depths and locations.

REPLY: We decided to leave figures 5 and 6 (current figures 7 and 8) because they emphasize the differences during periods characterized by high noise. Furthermore, we computed PPSD over the entire period of operation of the network (from 2016-04-01 to 2019-03-23) for each station, for each component, and for each sensor configuration. All the eight new figures (which include the 5$^{th}$, the 50$^{th}$, and the 95$^{th}$ percentiles according to your suggestion) are provided in the Supplement (Figures S1-S8). An additional Figure 10 with the comparison of the median values for sensors at different depths and locations (by considering the vertical channels) has been included in the revised manuscript. All the figures are commented and discussed in section 3.1.

Page 12, line 3 ". . .INS5 seismic station, where a small peak. . ." replace small with relevant. Indeed in figure 8 the H/V ratio exceeds 3, making this station the most amplified in the frequency band 2-5 Hz. Would have been interesting though to compute also spectral ratios among the stations having fixed one station as reference (e.g. INS4 or one station nearby free of site effects belonging the other permanent networks).

REPLY: You're right, we substituted "small" with "relevant". You're also right when you say that "Would have been interesting though to compute also spectral ratios among the stations having fixed one station as reference". Indeed, we would like in the next future to carry out a comprehensive study on site characterization in the study area by estimating also the velocity profile of the shallower layers in addition to SSR computations. With this data it will be possible to do several research activities, including this one; therefore, we added a sentence in the Discussion and conclusions section about the possibility to make such kind of research activity.

Page 14, line 1 Data availability section: this should evolve in a comprehensive description of the dataset availability. Provide details about where and how to access the data (fdsn web services at IRIS DMC). Data are available at IRIS DMC from 01.04.2016 to 25.06.2019. Is just this the open dataset described in this paper or this includes also open real-time data or periodic releases after certain embargo dates? Please specify in this section. A figure with the availability for the given period should be added. Would also be ideal to clearly state here if there is a license applied to the data and accordingly ask also IRIS DMC to include this in the DOI metadata of the network (including additional metadata as Funder, Sponsors, ORCIDs of the creators etc.). Moreover within the paper a seismic catalogue is mentioned and would be ideal to provide a link to it from here, either to an fdsn-event service or simply add catalogue to the supplementary material.

REPLY: Thank you for these fundamental hints. We extended Data availability section trying to provide a comprehensive information of the dataset according to your suggestions. We also clarified that data are released under the license CC BY 4.0. As mentioned above, Figure 4 of the revised version illustrates the data availability for the whole period of operation of each station of the seismic network. Upon our request, IRIS DMC updated the DOI metadata. As requested, the preliminary catalogue of seismicity has been provided as supplementary material (file "INSIEME-preliminary_catalogue.csv").

Page 14, lines 15 to page 1, line 3 Check this part carefully as most of this is redundant from the previous sections. Try to reduce redundancy and emphasize the part starting at page 15, line 4 stressing the peculiarities of this dataset.

REPLY: We tried to reduce redundancy and we added possible research activities which may be carry out with this dataset.

Page 14, line 26 "... detected 856 local natural and induced earthquakes ..." Can the earthquake catalogue, obtained from this network, be added to the supplementary material? Alternatively can the authors point to a repository where this catalogue is hosted?

REPLY: The preliminary catalogue of seismicity has been provided as supplementary material (file "INSIEME-preliminary_catalogue.csv").

**Reply to Supplementary material**

Being the dataset archived in a FDSN data centre providing standard data and metadata formats datalessSEED volumes can be omitted in my opinion. Within the manuscript the authors are referring to an own earthquake catalogue. This would be indeed a useful addition for the supplementary material. Or at least a link to an open standard fdsn web service where this can be obtained.

REPLY: We removed datalessSEED volumes from supplements and we added in the supplements the preliminary earthquake catalogue as requested.

[revised manuscript text omitted]
 ̶o̶f̶ ̶t̶h̶e̶ ̶I̶N̶S̶I̶E̶M̶E̶ ̶n̶e̶t̶w̶o̶r̶k̶. As evinced in Table 2, the broadband sensor of INS5 station was installed at 6 m depth during the whole period of observation; on the other hand, the broadband sensor of INS6 station was first placed on surface until 2017-03-22 and then moved in the shallow borehole at 6 m depth. Figure 9̶7̶ shows the comparison of spectrograms at the two stations over the whole investigated period. I̶t̶ ̶i̶s̶ ̶v̶e̶r̶y̶ ̶c̶l̶e̶a̶r̶ Tt̶he noise attenuation of about 20 dB at station INS5 with respect to station INS6 before 2017-03-22 is very clear, particularly a̶l̶o̶n̶g̶ for the two horizontal components, but the noise levels are comparable in the period of time when both t̶h̶e̶ stations have their r̶e̶s̶p̶e̶c̶t̶i̶v̶e̶ 
[revised manuscript text omitted]

---

## Author Response (AR2)

**Reply to Topical Editor (PAPER: The INSIEME seismic network: a research infrastructure for studying induced seismicity in the High Agri Valley (southern Italy)" by Tony Alfredo Stabile et al.)**

*2020-01-31*

**Topical Editor Comments to the Author:**

Dear Tony Alfredo Stabile,

many thanks for the careful revision of the paper and for addressing all reviewers comments. Before finally accepting your manuscript for publication, I would like to discuss with you the provision of the preliminary earthquake catalogue as supplementary material.

As signatory of the COPDESS Statement of Commitment and the Enabling FAIR Data Commitment Statement (https://copdess.org/) by Copernicus Publications, ESSD is not accepting data supplements. Instead we are requesting our authors to publish data via, ideally, domain) repositories. I could offer GFZ Data Services as suitable domain repository for solid earth sciences, if you need advice here. Once the earthquake catalogue has a valid DOI, please cite it in the manuscript and include it in the reference list.

many thanks for your understanding and best regards,

Kirsten Elger

**Reply to Topical Editor comments:**

Dear Kirsten Elger, Topical Editor,

we uploaded the preliminary earthquake catalogue file to Zenodo repository according to your request. The file has now a valid DOI (DOI:10.5281/zenodo.3632419) and it is available at the following link: https://zenodo.org/record/3632419#.XjQFIS2h0Wo

We also cited it in the manuscript and included it in the reference list. Thank you for your assistance.

Regards,

Tony Alfredo Stabile

[revised manuscript text omitted]